# Hierarchical Representations for Cross-task Automated Heuristic Design using LLMs

## Abstract

Designing heuristic algorithms for complex optimization problems is a time-consuming and expert-driven process. Recently, Automated Heuristic Design (AHD) using Large Language Models (LLMs) has shown significant promise for automating algorithm development. However, existing works mainly rely on programs to represent heuristics, which are inherently task-specific and fail to generalize as effectively as established metaheuristics like tabu search or guided local search. To bridge this gap, we introduce Multi-Task Hierarchical Search (MTHS), an LLM-guided evolutionary method that co-designs general-purpose metaheuristics and task-specific programs. MTHS employs a hierarchical representation and adopts a two-level evolution framework to evolve task-agnostic metaheuristics and task-specific program implementations simultaneously across multiple heuristic design tasks. During this evolution, a knowledge transfer mechanism allows learning from elite programs designed for other tasks. We evaluated MTHS on distinct combinatorial optimization problems, where it outperforms both commonly-used heuristics and existing LLM-driven AHD approaches. Our results demonstrate that the hierarchical representations facilitate effective multi-task AHD, and the evolved metaheuristics exhibit strong generalization to related tasks.

## 1 Introduction

Designing high-performance heuristic algorithms for complex problem-solving tasks is a notoriously challenging endeavor, traditionally relying on a time-consuming, expert-driven process of trial and error. Recently, Large Language Model (LLM)-driven Automated Heuristic Design (AHD) (Liu et al., 2024b; Ye et al., 2024; Zheng et al., 2025; Ye et al., 2025) has emerged as a powerful paradigm to automate algorithm development and mitigate this tedious process. This approach has already demonstrated its potential by automating the design of high-performance heuristics in diverse optimization domains including combinatorial optimization (Liu et al., 2024b; Ye et al., 2024), black-box optimization (van Stein & Bäck, 2024; Xie et al., 2025a), and Bayesian optimization (Yao et al., 2024).

A prevalent strategy in LLM-driven AHD is to embed LLMs as heuristic designers within iterative search frameworks (Zhang et al., 2024). Various search paradigms have been explored, from Evolutionary Computation (EC) (Liu et al., 2024b; Ye et al., 2024; Dat et al., 2025; Yao et al., 2025) to Monte Carlo Tree Search (MCTS) (Zheng et al., 2025). For instance, EoH (Liu et al., 2024b) evolves both natural language thoughts and executable code, ReEvo (Ye et al., 2024) integrates reflection strategies to refine the design process, and MCTS-AHD (Zheng et al., 2025) organizes heuristics in a tree to systematically explore the heuristic space.

However, a fundamental limitation persists in current AHD methods: they produce monolithic, task-specific heuristics. These approaches typically represent heuristics as either low-level programs (Zheng et al., 2025) or high-level thoughts (Liu et al., 2024b). Task-specific programs offer limited portability to new problems, while high-level thoughts are often too abstract to guarantee a direct correspondence with a high-performing implementation (Liu et al., 2024b). Consequently, existing systems must essentially restart the discovery process for each new problem, failing to institutionalize learning and generalize algorithmic knowledge across domains. This stands in stark contrast to human experts, who design and reuse metaheuristics, such as tabu search (Glover & Laguna, 1998) or simulated annealing (Van Laarhoven & Aarts, 1987), as general-purpose meta-

heuristics that are effective across a vast range of optimization tasks (Gendreau et al., 2010; Martí et al., 2025). While recent attempts have been made to enhance cross-distribution generalization (Shi et al., 2025; Liu et al., 2025), these methods are typically tailored for a single problem and do not generalize to others.

To bridge this gap, we argue that the key lies in creating hierarchical representations that separate general algorithmic logic from task-specific components, enabling cross-task automated heuristic design. We introduce the Multi-Task Hierarchical Search (MTHS), an LLM-guided hierarchical evolutionary framework designed to co-design general-purpose metaheuristics and their task-specific program implementations across multiple tasks simultaneously. MTHS leverages its hierarchical structure to explicitly transfer knowledge across tasks, allowing effective programs discovered in one task to inform and accelerate program design in others. Our primary contributions are threefold:

- We propose a hierarchical representation for LLM-driven AHD that consists of a task-agnostic metaheuristic and its task-specific program instantiations, thereby effectively enabling cross-problem generalization.
- We introduce the MTHS framework, which jointly designs the general metaheuristic and its task-specific implementations across diverse optimization tasks. At the high level, MTHS evolves metaheuristics; at the low level, it creates and refines programs and their associated key functions for each task. A cross-task knowledge transfer is adopted to learn from elite programs from other tasks.
- We conduct extensive experiments on four different combinatorial optimization problems, continuous black-box optimization problem, and admissible set problem. MTHS consistently discovers heuristics that outperform widely used heuristic baselines and state-of-the-art LLM-driven AHD methods. Crucially, the evolved metaheuristics exhibit strong generalization to related problems.

## 2 MULTI-TASK HIERARCHICAL SEARCH

### 2.1 HIERARCHICAL REPRESENTATION

This work addresses the problem of automated heuristic design across multiple, related tasks. The central goal is to discover high-level, general-purpose metaheuristics that can be specialized to achieve superior performance across multiple tasks. Formally, we are given a set of $m$ tasks, $\mathcal{T} = \{T_1, \ldots, T_m\}$. Each task $T_t$ is defined by a concise natural language description $D_t$, a program template $Temp_t$ providing the necessary inputs and outputs for execution, and a black-box evaluation function $E_t(\cdot)$ that returns a scalar performance score for a given program. Without loss of generality, we consider minimization problems in this paper.

Our representation for a candidate, which we term an *individual* $I_i$, is composed of two hierarchical levels: To be precise, each individual represents a complete metaheuristic for our multi-task AHD, encompassing both its high-level *metaheuristic* description and its task-specific program implementations. This structure is composed of two levels:

1. **Task-Agnostic Metaheuristic** ($MH_i$): At the highest level is a general-purpose *metaheuristic*, $MH_i$. Represented as a high-level algorithmic description, it captures the core problem-solving logic independent of any specific task.
2. **Task-Specific Programs** ($X_{i,t}$): For each task $T_t$, the metaheuristic $MH_i$ is instantiated into a concrete, executable *program*, $X_{i,t}$. This program adapts the general logic of $MH_i$ to the specific requirements of task $T_t$. Within each program $X_{i,t}$, we identify a performance-critical *key function*, denoted $F_{i,t}$.

The performance of an individual $I_i$ is evaluated based on the collective performance of its instantiated programs. Let $S_{i,t} = E_t(X_{i,t})$ be the score obtained by program $X_{i,t}$ on task $T_t$. The score list $\{S_{i,1}, \ldots, S_{i,m}\}$ is assigned to each individual $I_i$, which will be used in population management.

To ensure clarity, we will use the term *individual* to refer to this entire hierarchical entity and *metaheuristic* to refer specifically to the high-level description within it. While the distinction between "heuristic" and "metaheuristic" lacks a universal consensus in the literature (Gendreau et al., 2010;

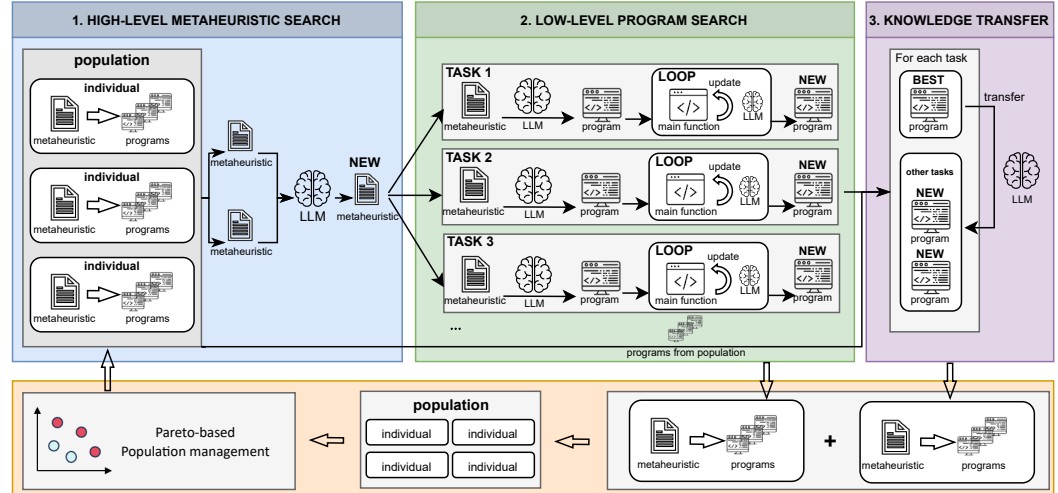

Figure 1: Overview of the MTHS pipeline. The pipeline consists of three main components: (1) High-level metaheuristic search, (2) Low-level program search, and (3) Knowledge transfer. In the high-level search, a population of metaheuristics is evolved, where each individual contains one metaheuristic paired with $m$ task-specific programs. For each newly generated metaheuristic, a low-level program search is performed for each task: a program is created for each task, and its key function is identified and refined using LLMs. This produces a new candidate comprising the metaheuristic and its $m$ programs. Next, a knowledge transfer phase is applied for each task: the best-performing program across both the existing population and the new candidate is identified and used to update the other $m$-1 programs within the same metaheuristic. Candidates produced from both low-level search and knowledge transfer are added to the population. Finally, a Pareto-based population management step selects individuals to form the next generation.

Martí et al., 2025), we adopt the view that metaheuristics represent a more general problem-solving paradigm. Nevertheless, given their conceptual overlap, we may use these terms interchangeably where the context allows.

## 2.2 FRAMEWORK

We introduce Multi-Task Heuristic Search (MTHS), a framework that automates heuristic design across a set of related tasks using a two-level evolutionary algorithm (see Figure 1 and Algorithm 1). The process begins by prompting LLMs with descriptions of all tasks to seed an initial high-level population ($\mathcal{P}_H$) of diverse, task-agnostic metaheuristics. Each metaheuristic represents a general problem-solving strategy intended to be effective across multiple tasks. For each of these metaheuristics, MTHS initiates a distinct low-level search for every individual task. This low-level process evolves separate populations ($\mathcal{P}_{L,t}$) of task-specific programs. A knowledge transfer mechanism then shares insights from the best-performing programs across tasks. This hierarchical structure enables MTHS to simultaneously conduct broad strategic exploration at the shared metaheuristic level and specialized, fine-grained program optimization at the individual task level. We introduce each phase as follows. We expand the subalgorithms and present the detailed specific prompts in Appendix B.

## 2.3 HIGH-LEVEL EVOLUTION

The high-level evolution maintains a population of individuals, $\mathcal{P}_H$. In each generation, we employ the LLM as an evolutionary operator to generate a new candidate metaheuristic. The process begins by selecting a set of $k$ parent individuals, $\{I_1, \ldots, I_k\}$, from $\mathcal{P}_H$. The corresponding metaheuristic descriptions of these parents, $\{MH_1, \ldots, MH_k\}$, are then formatted into a carefully designed prompt. This prompt instructs the LLM, $\mathcal{L}$, to synthesize a novel and potentially superior meta-

---

**Algorithm 1** Multi-Task Heuristic Search (MTHS)

---

**Input:**
1: $\mathcal{T} = \{T_1, \ldots, T_m\}$: Set of $m$ tasks, each with description $D_t$, template $Temp_t$, and evaluator $E_t(\cdot)$
2: $\mathcal{L}$: Large Language Model
3: $N_{eval}$: total evaluation limits
4: $N_H, k, N_L$: High-level population size and number of parents, Low-level evaluation budget
**Output:** The final population of high-performing individuals $\mathcal{P}_H$
5: **procedure** MTHS($\mathcal{T}, \mathcal{L}, N_{eval}, N_H, k$)
6:    $\mathcal{P}_H \leftarrow \emptyset$
7:    InitialMHs $\leftarrow \mathcal{L}($BuildInitialPrompt$(\{D_t\}_{t=1}^m))$
8:    **for** each $MH_{init}$ in InitialMHs **do**
9:        $I_{new} \leftarrow$ LowLevelEvolution($MH_{init}, \mathcal{T}, N_L, \mathcal{L}$)
10:        $\mathcal{P}_H \leftarrow \mathcal{P}_H \cup \{I_{new}\}$
11:    **while** evaluation count $\leq N_{eval}$ **do**
12:        $\{I_1, \ldots, I_k\} \leftarrow$ SelectParents($\mathcal{P}_H, k$)
13:        $\{MH_1, \ldots, MH_k\} \leftarrow \{I_1, \ldots, I_k\}$
14:        prompt $\leftarrow$ BuildEvolutionPrompt($\{D_t\}_{t=1}^m, \{MH_j\}_{j=1}^k$)
15:        $MH_{new} \leftarrow \mathcal{L}($prompt$)$
16:        $I_{new} \leftarrow$ LowLevelEvolution($MH_{new}, \mathcal{T}, N_L, \mathcal{L}$)     ▷ *Sec. 2.4*
17:        **if** $I_{new}$ is valid **then**
18:            $I_{new} \leftarrow$ KnowledgeTransfer($I_{new}, \mathcal{T}, \mathcal{L}$)     ▷ *Sec. 2.5*
19:            $\mathcal{P}_H \leftarrow$ UpdatePopulation($\mathcal{P}_H \cup \{I_{new}\}, N_H$)     ▷ *Sec. 2.6*
20:    **return** $\mathcal{P}_H$

21: **procedure** LOWLEVELEVOLUTION($MH_{new}, \mathcal{T}, N_L, \mathcal{L}$)
22:    $I_{new} \leftarrow$ new Individual with $MH_{new}$
23:    **for** $t \leftarrow 1$ **to** $m$ **do**
24:        $X_{new,t} \leftarrow \mathcal{L}($BuildProgramPrompt($MH_{new}, T_t.Temp, \mathcal{L}$))
25:        $F_{new,t} \leftarrow \mathcal{L}($BuildKeyFuncPrompt($T_t.D, X_{new,t}, \mathcal{L}$))
26:        $(X_{new,t}^*, S_{new,t}^*) \leftarrow$ EvolveKeyFunction($X_{new,t}, F_{new,t}, T_t.E, \mathcal{L}$)
27:        $I_{new}.X_{new,t} \leftarrow X_{new,t}^*$
28:        $I_{new}.S_{new,t} \leftarrow S_{new,t}^*$
29:    **return** $I_{new}$

---

heuristic, adhering to a predefined description template. The LLM's textual output constitutes the metaheuristic description, $MH_{new}$, for the new offspring.

## 2.4 LOW-LEVEL EVOLUTION

Each newly generated metaheuristic $MH_{new}$ must be instantiated and optimized for all $m$ tasks to determine its fitness. This evaluation is a multi-step procedure executed for each task $T_t$. First, in **i) Task-Specific Program Generation**, the LLM generates a full, compilable program $X_{new,t}$ by integrating the logic of $MH_{new}$ with the task-specific template $Temp_t$. Second, during **ii) Key Function Identification**, the LLM is prompted to analyze the generated code $X_{new,t}$ and identify its most performance-critical component, which we designate as the key function $F_{new,t}$. Third, a dedicated low-level evolutionary search is performed to refine the key function in a process of **iii) Key Function Refinement**. An ephemeral population $\mathcal{P}_{L,t}$ is initialized with variants of $F_{new,t}$ generated by the LLM. This population then undergoes a short evolutionary process for a fixed budget of $N_L$ evaluations. The LLM acts as a mutation operator, creating new function variants from existing high-performing ones. Each new variant is injected back into the base program $X_{new,t}$ and evaluated using $E_t(\cdot)$. Finally, for **iv) Fitness Assignment**, after the low-level search concludes, the best-performing key function variant, $F_{new,t}^*$, is identified. The program incorporating this optimized function, $X_{new,t}^*$, yields the fitness score $S_{new,t}$ for the metaheuristic $MH_{new}$ on task $T_t$.

Once this process is completed for all $m$ tasks, a new individual $I_{new}$ is formed, comprising the metaheuristic $MH_{new}$, its vector of scores $\{S_{new,1}, \ldots, S_{new,m}\}$, and the set of optimized programs $\{X^*_{new,1}, \ldots, X^*_{new,m}\}$. This individual is then added to the high-level population $\mathcal{P}_H$.

## 2.5 Knowledge Transfer

To facilitate explicit cross-task learning, we introduce a knowledge transfer phase. For the new individual $I_{new}$, we identify its best-performing program, $X^*_{new,src}$, on some source task $T_{src}$. The LLM is then prompted to adapt the logic of this program to every other target task $T_{tgt}$ (where $tgt \neq src$). This adaptation creates a new set of candidate programs. If an adapted program for $T_{tgt}$ achieves a better score than the incumbent program $X^*_{new,tgt}$, it replaces it. This process directly transfers successful algorithmic patterns discovered on one task to others within the context of the same metaheuristic, $MH_{new}$.

## 2.6 Pareto-based Population Management

MTHS uses a Pareto-based survival strategy to manage the high-level population $\mathcal{P}_H$. Since each metaheuristic is evaluated on $m$ tasks, its performance is represented by a score vector $\mathbf{S}_i = (S_{i,1}, \ldots, S_{i,m})$, framing the search as a multi-objective optimization problem. The most straightforward way to tackle multi-objective search is to transfer the multiple objectives into a single objective using some scalarization method, such as weighted-sum. However, it is hard to determine proper weights because the task scores are on different scales.

Therefore, we adopt a Pareto-based approach that works as follows: **i): Task Champions (Elitism):** For each task $t \in \{1, \ldots, m\}$, the individual with the highest score $S_{i,t}$ on that task is automatically preserved for the next generation. This ensures that the best-known performance on any single task is never lost. **ii): Pareto Dominance Ranking:** All remaining individuals in the candidate pool are ranked based on Pareto dominance. An individual $I_i$ is said to dominate $I_j$ if: $(\forall t,\ S_{i,t} \geq S_{j,t}) \wedge (\exists t',\ S_{i,t'} > S_{j,t'})$, where missing scores are treated as $-\infty$. Individuals are sorted into non-dominated fronts. **iii): Selection and Truncation:** The next generation is populated by adding individuals from the first non-dominated front, then the second, and so on, until the population size $N_H$ is reached. If adding an entire front would exceed the population size, individuals from that front are selected based on their average scores on all tasks. An illustration of populations in objective space is presented in Appendix E.

# 3 Experimental Studies

## 3.1 Tasks and Datasets

We evaluate our method on four combinatorial optimization problems: the Traveling Salesman Problem (TSP), Capacitated Vehicle Routing Problem (CVRP), Flow Shop Scheduling Problem (FSSP), and Bin Packing Problem (BPP). For each problem, we generate a set of 64 diverse instances for the heuristic evolution phase. The final performance of the evolved heuristics is then validated on established, standard benchmark datasets. Further details on instance generation and benchmark specifics are provided in Appendix C.

- **Traveling Salesman Problem (TSP):** We aim to find the shortest tour visiting a set of locations. Our evolution set consists of 100node instances with locations uniformly sampled in $[0, 1]^2$. Fitness is the average optimality gap relative to the Concorde solver (Applegate et al., 2006). For final evaluation, we use standard instances from TSPLib (Reinelt, 1991).
- **Capacitated Vehicle Routing Problem (CVRP):** The goal is to design minimum-cost routes for a fleet of capacitated vehicles to serve a set of customers. The evolution set contains 100-customer instances. Fitness is measured as the average gap to solutions found by the LKH3 solver (Helsgaun, 2017). We test the final heuristics on CVRPLib benchmarks (Uchoa et al., 2017).
- **Flow Shop Scheduling Problem (FSSP):** We seek to schedule $n$ jobs on $m$ machines to minimize the makespan (total completion time). Our evolution instances feature 50 jobs and a variable number of machines ($m \in [2, 20]$). Fitness is the average makespan. Final validation is performed on the Taillard benchmark suite (Taillard, 1993).

Table 1: Results on standard benchmark instances from TSPLib and CVRPLib (Sets A, B, E, F, M, P, and X). The table reports the average percentage gap to the best-known solutions for four distinct groups of heuristics: constructive heuristics, metaheuristics, LLM-designed heuristics, and our methods. The best result for each instance set is highlighted in **bold**, and the second-best one is underlined. For comparison, we also include the SOTA solvers LKH3 (for TSP) (Helsgaun, 2017) and HGS (Vidal, 2022).

| | TSPLib | | | | CVRPLib | | | | | | |
|---|---|---|---|---|---|---|---|---|---|---|---|
| | 50-99 | 100-199 | 200-499 | 500-1000 | A | B | E | F | M | P | X |
| LKH3 | 0.49 | 0.12 | 0.00 | 0.15 | - | - | - | - | - | - | - |
| HGS | - | - | - | - | 0.32 | 0.36 | 0.10 | 0.72 | 1.02 | 0.25 | 0.59 |
| NN | 27.07 | 23.76 | 24.79 | 26.57 | 39.40 | 42.32 | 41.51 | 60.01 | 52.88 | 36.18 | 27.63 |
| Insert | 13.99 | 16.15 | 20.00 | 26.30 | 33.86 | 33.07 | 32.51 | 65.45 | 44.49 | 25.96 | 31.19 |
| Or-tools SA | 3.01 | 3.74 | 4.62 | 10.08 | 6.58 | 5.40 | 7.48 | 9.44 | 15.54 | 5.60 | 7.82 |
| Or-tools TS | 1.81 | 3.20 | 4.62 | 10.11 | **1.05** | **1.12** | 1.57 | 4.56 | 5.74 | 1.11 | 6.06 |
| Or-tools GLS | 0.63 | 1.62 | 3.34 | 6.84 | 1.24 | 1.14 | 1.30 | 3.49 | 7.74 | 1.07 | 6.29 |
| MS | 1.82 | 2.92 | 4.15 | 6.58 | 8.19 | 11.60 | 10.25 | 9.65 | 42.66 | 7.52 | 42.85 |
| ALNS | 1.62 | 1.90 | 5.24 | 8.28 | 6.39 | 5.80 | 3.93 | 3.56 | 14.94 | 4.61 | 11.33 |
| TS | 4.10 | 5.54 | 7.52 | 12.61 | 5.06 | 4.00 | 5.83 | 4.51 | 6.40 | 5.56 | 5.77 |
| ACO_EoH | 7.95 | 8.09 | 14.71 | 22.36 | 20.11 | 16.05 | 18.15 | 34.48 | 31.20 | 12.90 | 19.72 |
| ACO_MCTS | 3.68 | 3.40 | 9.13 | 22.64 | 15.70 | 10.90 | 17.80 | 35.53 | 29.34 | 12.82 | 18.77 |
| GLS_EoH | 0.67 | 0.63 | 1.62 | 2.67 | 2.69 | 3.89 | 3.99 | 6.56 | 4.43 | 5.23 | 5.17 |
| GLS_ReEvo | 0.79 | 0.68 | 1.71 | 2.72 | 2.60 | 3.72 | 4.00 | 6.96 | **2.45** | 5.61 | 5.62 |
| GLS_MCTS | 0.75 | 0.64 | 1.53 | 2.93 | 3.07 | 3.97 | 4.79 | 6.89 | 4.23 | 5.02 | 6.22 |
| STHS | 0.87 | 0.60 | 1.47 | 3.59 | 3.48 | 3.88 | 4.41 | 3.41 | 6.80 | 5.64 | 5.36 |
| MTHS | 0.72 | **0.49** | **1.03** | **2.64** | 1.08 | 1.50 | **0.94** | **1.23** | 3.51 | **1.06** | **4.29** |

- **Bin Packing Problem (BPP):** The objective is to pack items of various sizes into the minimum number of fixed-capacity bins. Following prior work (Ye et al., 2024; Zheng et al., 2025), our evolution instances feature a bin capacity of 150 and item sizes sampled from $[20, 100]$. Fitness is the average number of bins used.
- **Black-Box Optimization (BBO):** The objective is to find a vector $x^*$ that minimizes a function $f(x)$ whose analytical form is unknown. Our evaluation instances are five standard 20-dimensional benchmark functions (Sphere, Rosenbrock, Rastrigin, Ackley, Griewank) with varying search domains. Performance is measured by how close the solver gets to the known global minimum of 0.0 for each function.
- **Admissible Set Problem (ASP):** The goal is to construct the largest possible set of vectors satisfying specific combinatorial constraints, avoiding predefined "bad triples". Our evaluation instances are standard benchmarks defined by vector dimension and weight pairs: $(n = 15, w = 10)$, $(n = 12, w = 7)$, $(n = 21, w = 15)$, and $(n = 24, w = 17)$. The quality of the solution is measured by the size of the final admissible set generated.

## 3.2 METHODS AND SETTINGS

We compare our method, MTHS, against a diverse set of baselines representing the state of the art in both conventional and LLM-assisted heuristic design.

- **Conventional Heuristics:** We include widely-used constructive: Nearest Neighbor (**NN**) (Rosenkrantz et al., 1977) and a standard Insertion heuristic (**Insert**) (Rosenkrantz et al., 1977) and metaheuristic: Tabu Search (**TS**) (Glover & Laguna, 1998), Adaptive Large Neighborhood Search (**ALNS**) (Pisinger & Ropke, 2018), Memetic Search (**MS**) (Neri et al., 2011), and Guided Local Search (**GLS**) (Voudouris et al., 2010). For FSSP, we investgate **GUPTA** (Gupta, 1971),**CDS** (Campbell et al., 1970), **NEH** (Nawaz et al., 1983) and **NEHFF** (Fernandez-Viagas & Framinan, 2014), where NEH (Nawaz et al., 1983) and NEHFF (Fernandez-Viagas & Framinan, 2014) are widely recognized heuristics for this problem.
- **Google OR-Tools:** A high-performance, unified solver for CO problems. We utilize its standard metaheuristic solvers: Guided Local Searach (**OR-Tools GLS**), Simulated Annealing (**OR-Tools**

Table 2: Results on benchmark FSSP instances. The average gap (%) to the upper bounds from Taillard's FSSP benchmarks (Taillard, 1993), calculated over the 10 instances in each problem set. A set with $n$ jobs and $m$ machines is denoted as $n\_m$. The best result in each row is shown in **bold**, and the second-best is underlined.

| | 20_5 | 20_10 | 20_20 | 50_5 | 50_10 | 50_20 | 100_5 | 100_10 | 100_20 | Average |
|---|---|---|---|---|---|---|---|---|---|---|
| GUPTA | 12.89 | 23.42 | 21.79 | 12.23 | 20.11 | 22.78 | 5.98 | 15.03 | 21.00 | 17.25 |
| CDS | 9.03 | 12.87 | 10.35 | 6.98 | 12.72 | 15.03 | 5.10 | 9.36 | 13.55 | 10.55 |
| NEH | 3.24 | 4.05 | 3.06 | 0.57 | 3.47 | 5.48 | 0.39 | 2.07 | 3.58 | 2.88 |
| NEHFF | 2.30 | 4.15 | 2.72 | 0.40 | 3.62 | 5.10 | 0.31 | 1.88 | 3.73 | 2.69 |
| LS | 1.91 | 2.77 | 2.60 | 0.32 | 3.33 | 4.67 | 0.28 | 1.38 | 3.51 | 2.31 |
| ILS | 0.18 | 0.59 | 0.45 | 0.03 | 1.27 | 1.99 | **-0.03** | 0.34 | 1.29 | 0.68 |
| MTHS | **-0.01** | **0.03** | **0.03** | **0.00** | **0.22** | **0.45** | -0.02 | **0.52** | **0.98** | **0.24** |

**SA**) (Van Laarhoven & Aarts, 1987), and Tabu Search (**OR-Tools TS**) with their default parameter configurations.

- **LLM-driven Methods:** We compare against three recent LLM-based AHD methods: **EoH** (Liu et al., 2024b), **ReEvo** (Ye et al., 2024), and **MCTS-AHD** (Zheng et al., 2025). As these methods operate on a base heuristic framework, we test them with Ant Colony Optimization (ACO) and GLS, consistent with their original papers.

- **MTHS (Ours):** We evaluate our method in two configurations: **MTHS (Multi-Task):** The full proposed method, and **STHS (Single-Task):** An ablation where knowledge transfer and Pareto-based population management are disabled to assess the single-task performance of our hierarchical search.

**Experimental Setup for LLM-driven AHD**  For MTHS, we conduct AHD on three tasks (i.e., TSP, CVRP and FSSP) with a budget of 1,000 program evaluations (i.e., $N_{eval} = 1,000$). The high-level population size is $N_H = 8$ and the low-level search budget is $N_L = 4$. For STHS and all compared LLM-driven AHD methods, including EoH, ReEvo, and MCTS-AHD, we conduct one search run per task with a budget of 1,000 program evaluations with their default settings. To prevent excessively long evaluations from stalling the search process, we impose a 20-minute time limit on each individual heuristic evaluation. We used GPT-5-mini as the underlying LLM for both our method (MTHS) and the three AHD baselines: EoH, ReEvo, and MCTS with GLS. For the baselines that use an ACO framework, we directly adopted the best heuristics reported by Zheng et al. (2025) rather than re-running the search. It has been demonstrated that GLS outperforms ACO Zheng et al. (2025). A summary of settings and running times is listed in Appendix D.

**Implementation and Execution Environment**  All heuristic algorithms were implemented in Python, with the exception of Google OR-Tools, which uses a C++ library with a Python interface. Following standard practice in LLM-driven AHD research, we use the Numba JIT compiler to accelerate computationally intensive components, such as local search operators, for all metaheuristics, including those designed by the LLM-based methods.

Establishing a perfectly fair comparison based on a fixed evaluation budget is challenging due to the diverse frameworks and iterative components of different metaheuristics. Therefore, we adopted a time-based comparison protocol. We carefully configured the parameters of all baseline methods to commonly accepted values, ensuring that all algorithms had a comparable average wall-clock time for their execution. Detailed parameter settings are provided in the Appendix D.

The LLM-driven AHD experiments were conducted on a workstation equipped with two Intel Xeon 6248R CPUs and 128 GB of RAM. A single multi-task AHD using MTHS, utilizing 8-core parallel evaluations, took approximately 1.5 days to complete. The final heuristic evaluations were performed on a machine with an Intel Core Ultra 7 CPU and 32 GB of RAM.

## 3.3 MAIN RESULTS

We present our main experimental results in Table 1 and Table 2. A key contribution of our work is the ability of MTHS to discover a general-purpose metaheuristic. To demonstrate this, we selected a

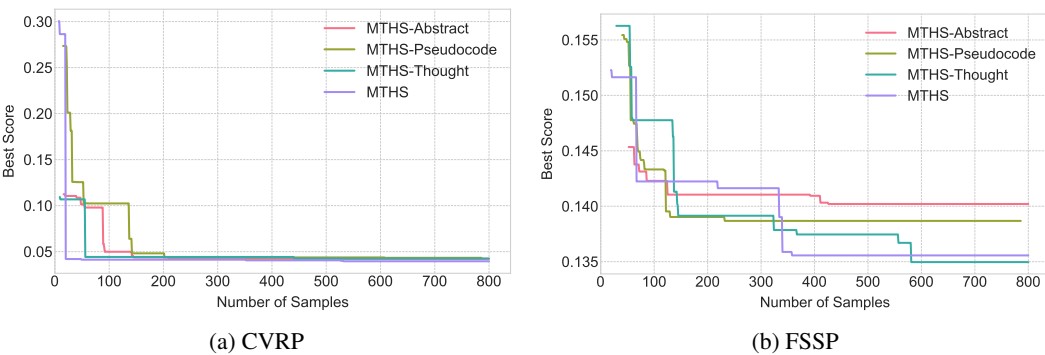

(a) CVRP

(b) FSSP

Figure 2: A comparison of different metaheuristic representations on CVRP and FSSP.

single metaheuristic from the final MTHS population and applied its associated three programs to the three distinct problem domains. This approach highlights the task-agnostic nature and generalization capabilities of the designed metaheuristic across different tasks. In contrast, existing LLM-driven AHD frameworks, including EoH, ReEvo, and MCTS-AHD, must execute a separate search to design a specialized heuristic for each task. The performance metric is the percentage gap relative to best-known solutions (for TSP and CVRP) or established upper bounds (for FSSP), with lower values signifying superior performance. For clarity, the best-performing heuristic is marked **in bold**, while the second-best is underlined.

Table 1 summarizes the results on the TSPLib and CVRPLib benchmarks. Our proposed method, MTHS, demonstrates superior performance, consistently outperforming all baseline methods across nearly all instance sets. For the TSP, MTHS achieves the lowest average optimality gaps on three size categories, from smaller instances to the largest ones (500-1000 nodes). The performance advantage of MTHS is consistent on the CVRP benchmarks. It secures the best results on four out of the seven CVRPLib sets (E, F, P, X) and is highly competitive on the remaining three. In contrast, while highly optimized solvers like Google OR-Tools perform well, especially on smaller CVRP instances, their performance degrades on larger TSP instances compared to the best LLM-evolved heuristics. Furthermore, comparing MTHS to its single-task ablation, STHS, reveals the clear benefit of multi-task learning; MTHS consistently outperforms STHS, underscoring the effectiveness of knowledge transfer in discovering more robust and powerful heuristics.

Table 2 shows the results on the Taillard benchmark for FSSP. The heuristic discovered by MTHS establishes a new state of the art, substantially outperforming all conventional constructive heuristics and local search methods. It achieves an average gap of just 0.24%. On several instance sets (20_5 and 100_5), the MTHS-designed heuristic finds solutions that are slightly better than or close to the existing upper bounds provided in Taillard (1993).

## 3.4 METAHEURISTIC REPRESENTATION

We now analyze the representation MTHS uses to design metaheuristics, a key part of its success. The representation determines the LLM's level of abstraction, which in turn affects search efficiency and the quality of the resulting algorithms.

We compare four distinct metaheuristic representation strategies. Examples of different metaheuristic representations are provided in Appendix E.

- **Abstract:** The LLM is prompted to design a task-agnostic code structure directly, without a predefined template. This offers maximum flexibility but minimal structural guidance.
- **Pseudocode:** The LLM is prompted to design a task-agnostic pseudocode, which is then translated into executable code.
- **Thought:** The LLM describes the high-level strategy or "thought proces" of a metaheuristic.
- **MTHS (Template):** Our proposed method, which uses a structured, task-agnostic template to define the metaheuristic's components and control flow.

We use different metaheuristic representation in MTHS and perform the cross-task AHD on the three tasks with the same settings. Figure 2 illustrates the convergence behaviour of the automated search process for each representation on the CVRP and FSSP. It depicts the current best score (related gap to baseline on training instances) with respect to the number of program samples. The results clearly demonstrate the superiority of the template-based metaheuristic representation used in MTHS. For both problems, MTHS achieves a faster convergence compared to the other representations. This suggests that providing the LLM with a well-defined, modular structure is helpful for efficiently navigating the vast search space of possible metaheuristics.

### 3.5 GENERALIZATION TO NEW TASKS AND LLMS

A central hypothesis of our work is that a well-designed, task-agnostic metaheuristic can serve as a powerful and generalizable scaffold for solving novel problems. To test this, we evaluate the generalization of a metaheuristic discovered by MTHS on a diverse set of unseen tasks and across different LLMs. We investigate three distinct problems: the Bin Packing Problem (BPP), a Black-Box Optimization Problem (BBOP), and an Admissible Set Problem (ASP). The BPP is a combinatorial optimization task, similar in nature to the problems used to train MTHS. In contrast, the BBOP is a continuous optimization problem, while the ASP represents a less relative task domain. Our results demonstrate that the MTHS-designed metaheuristic generalizes effectively to BPP and BBOP, but shows no significant improvement for ASP. We present detailed results for the BPP in the main text and provide the findings for the other two problems in the Appendix.

Specifically, we prompt LLMs to generate code for solving BPP without any evolution (i.e., repeated sampling). We evaluate three models, including GPT-5-mini, Gemini-2.5-pro, and Claude-3.7-Sonnet, under two conditions: i) the model writes a solver from scratch, and ii) the model is explicitly instructed to implement a solver based on the metaheuristic template designed by MTHS (+MH). For each model and condition, we generate 100 programs and evaluate their performance on five BPP instances.

Figure 3 presents the performance distribution of the top 10 programs from each setting. The results show a notable and consistent improvement when the LLMs are guided by the MTHS-designed metaheuristic. For all three models, the + MH setting yields programs with significantly lower optimality gaps. Notably, Gemini-2.5-pro, when guided by the metaheuristic, produced a solver achieving a near-optimal gap of 0.002%, while when no metaheuristic is given, it struggled in designing high-quality BPP solvers. Results demonstrate that the task-agnostic metaheuristic designed by MTHS provides a general problem-solving logic that effectively transfers to new related tasks and can be leveraged by different LLMs.

To further validate the effectiveness of this generalization, we evaluate the top-performing LLM-generated solvers against established baselines on two BPP test sets with 500 and 1000 items. Each set contains 64 instances, and the average gap to the lower bound is reported. The solver generated by Gemini-2.5-pro with our metaheuristic guidance (+ MH) achieves state-of-the-art performance, recording optimality gaps of just 0.34% and 0.25% on the n=500 and n=1000 instances, respectively. This significantly outperforms not only the scratch-generated LLM solvers but also existing task-specific approaches like MCTS-AHD (0.48% and 0.53%). This demonstrates that the MTHS-discovered metaheuristic can generalized to other related tasks and enables LLMs to create programs that are not only conceptually sound but also highly competitive.

## 4 CONCLUSION

This paper addresses the limited cross-task generalization of current task-specific LLM-dirven AHD. We introduced Multi Task Hierarchical Search (MTHS), a framework that shifts the focus from crafting monolithic solvers to co-designing task-agnostic metaheuristics together with their task-specific realizations. Through a hierarchical representation and evolution, the method creates high-level metaheuristics that are reusable across tasks. Experiments on four problems show that meta-heuristics produced by our approach outperform strong classical baselines, specialized metaheuristic solvers, and existing LLM-driven AHD methods. More importantly, the learned metaheuristic exhibits strong out-of-distribution behaviour. Used as a template on an unseen BPP, it enabled different LLMs to instantiate high-quality solvers without iterative search. These results indicate that design-

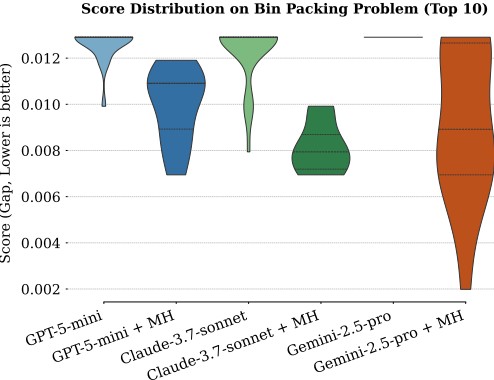

Figure 3: A comparison of results on BPP instances. + MH represents that we inform LLM to create programs using a given metaheuristic automatically designed by MTHS.

Table 3: Results on two sets of BPP instances. The results of three baseline methods are from (Zheng et al., 2025). + MH represents that we inform LLM to create programs using a given metaheuristic automatically designed by MTHS.

| Method | n500, c150 | n1000, c150 |
|---|---|---|
| EoH | 0.75% | 0.85% |
| ReEvo | 1.76% | 2.06% |
| MCTS-AHD | 0.48% | 0.53% |
| GPT-5-mini | 0.98% | 0.98% |
| GPT-5-mini + MH | 0.82% | 0.65% |
| Gemini-2.5-pro | 1.32% | 1.25% |
| Gemini-2.5-pro + MH | **0.34%** | **0.25%** |

ing at the metaheuristic level within a hierarchical representation offers a viable path to cross-task generalization in LLM-driven automated algorithm design.

In future work, we plan to expand the task suite, refine transfer mechanisms, and incorporate resource and reliability constraints directly into the search process. A deeper analysis of when and why transfer succeeds could further amplify the benefits of this paradigm.

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

# SUPPLEMENTARY MATERIAL

## TABLE OF CONTENTS

# A    RELATED WORKS

## A.1    AUTOMATED HEURISTIC DESIGN (AHD)

Automated Heuristic Design (AHD), often discussed under the umbrella of hyper-heuristics (Burke et al., 2018; Stützle & López-Ibáñez, 2018), aims to automate the process of selecting, combining, or generating simpler heuristics to solve complex computational search problems (Pillay & Qu, 2018). AHD methods are broadly categorized into selection and generation approaches.

Genetic programming or grammatical evolution (O'Neill & Ryan, 2001) are commonly used in generating new algorithms from fundamental building blocks. Recent advances in this area include component-based frameworks that assemble novel algorithms by integrating diverse operators and algorithmic stages (Bezerra et al., 2015; Qu et al., 2020). While powerful, these approaches often rely on hand-crafted components and require significant domain-specific knowledge, which can limit their flexibility and ease of application.

## A.2    LLM-DRIVEN AHD

The advent of LLMs has introduced a new paradigm for AHD. A prominent strategy employs an evolutionary framework where LLMs iteratively propose and refine algorithms (Zhang et al., 2024; van Stein & Bäck, 2024). For example, Evolution of Heuristics (EoH) (Liu et al., 2024b) evolves both high-level thoughts and executable code using distinct prompt strategies to guide the search. FunSearch (Romera-Paredes et al., 2024) uses a multi-island evolutionary approach with a focused prompt strategy for refinement, while ReEvo (Ye et al., 2024) integrates reflection mechanisms to provide LLMs with more structured guidance. Other search strategies, such as Monte Carlo Tree Search (MCTS) (Zheng et al., 2025) and neighborhood search (Xie et al., 2025b), have also been explored to steer the design process.

Despite their success, a common limitation of these methods is their focus on discovering a single heuristic optimized for average performance on a specific task. The heuristic and knowledge can hardly be generalized to solving other tasks.

## A.3    MULTI-TASK LEARNING FOR AHD

In the adjacent field of neural combinatorial optimization, multi-task learning has emerged as a key strategy for improving cross-problem generalization (Liu et al., 2024a; Berto et al., 2024). Researchers have developed single neural solvers trained across multiple problem types. Others demonstrate that models pre-trained on one problem (e.g., TSP) can be efficiently fine-tuned for related tasks (e.g., VRPs) using techniques like LoRA (Lin et al., 2024). Moverover, recent work (Shi et al., 2025) has explored using LLMs to extract symbolic features that enhance the generalization of a backbone neural solver. However, these neural solvers are often black-box models that lack interpretability and typically require large datasets and substantial computational resources for training.

## A.4    HEURISTIC REPRESENTATION IN LLM-DRIVEN AHD

The representation of the heuristic itself is a critical design choice in LLM-driven AHD. A common approach, popularized by EoH (Liu et al., 2024b), is a dual "thought-and-code" representation, where a high-level idea guides the generation of executable code. This or single code-based representations have been adopted by many subsequent works (Ye et al., 2024; Zheng et al., 2025; van Stein & Bäck, 2024). Recent explorations have introduced intermediate representations like pseudocode (Gurkan et al., 2025) or more flexible code structures (Novikov et al., 2025).

Closer to our work, some methods have used high-level algorithmic templates to enable meta-learning across different distributions of the same problem (Shi et al., 2025). However, to our knowledge, the challenge of learning generalizable metaheuristic structures that can be applied across entirely different tasks has not yet been addressed. Our hierarchical representation is designed specifically to fill this gap.

# B  MORE METHOD DETAILS

## B.1  HIGH-LEVEL POPULATION INITIALIZATION

The `InitializePopulation` procedure (Algorithm 2) is responsible for seeding the initial high-level population, $\mathcal{P}_H$, which serves as the starting point for the main evolutionary search. The goal is to generate a diverse and competent set of initial individuals, where each individual represents a complete multi-task problem-solving strategy.

The procedure begins by constructing a single, comprehensive prompt using the `BuildInitialPrompt` function. This prompt aggregates the descriptions, $\{D_t\}_{t=1}^m$, of all $m$ tasks in the set $\mathcal{T}$. This contextual information guides the LLMs ($\mathcal{L}$) to generate a set of initial metaheuristics, denoted as `InitialMHs`. Each metaheuristic, $MH_{init}$, is a high-level textual description of a problem-solving approach.

For each generated $MH_{init}$, the procedure invokes `LowLevelEvolution` (as defined in the main MTHS algorithm). This critical step translates the abstract metaheuristic into a concrete, executable individual, $I_{new}$. The `LowLevelEvolution` procedure instantiates the metaheuristic into task-specific programs, refines them, and evaluates their performance, consuming a low-level evaluation budget of $N_L$. The resulting individual, $I_{new}$, contains a collection of optimized programs $\{X_{new,t}^*\}_{t=1}^m$ and their corresponding scores $\{S_{new,t}^*\}_{t=1}^m$.

If the newly created individual $I_{new}$ is deemed valid (e.g., it compiles and runs without fatal errors), it is added to the high-level population $\mathcal{P}_H$. This process repeats until the population reaches its target size, $|\mathcal{P}_H|$. The final, fully populated $\mathcal{P}_H$ is then returned, ready for the main evolutionary loop of the MTHS algorithm.

## B.2  LOW-LEVEL KEY FUNCTION EVOLUTION

The `EvolveKeyFunction` procedure (Algorithm 3) implements a fine-grained, task-specific optimization process. It is a core component of the `LowLevelEvolution` routine and is responsible for refining a single program by iteratively improving its most critical component: the key function. Its inputs are the initial program code $X_t$ for a task $T_t$, the identified key function $F_t$ within that code, the task object $T_t$ (which provides the description $D_t$ and evaluator $E_t(\cdot)$), the LLM $\mathcal{L}$, and the low-level evaluation budget $N_L$.

The procedure operates as a micro-evolutionary search. It first initializes a local, low-level population, $\mathcal{P}_{L,t}$, by seeding it with the initial program $X_t$, its key function $F_t$, and its evaluated score. The main loop then commences, running until the evaluation budget $N_L$ is exhausted.

In each iteration, a parent program, $p_{parent}$, is selected from $\mathcal{P}_{L,t}$ using a selection strategy (e.g., tournament selection). A mutation prompt is then constructed via `BuildMutationPrompt`, providing the LLM with the task description $T_t.D$ and the body of the parent's key function, $p_{parent}.$function. The LLM acts as a sophisticated mutation operator, generating a new function body, $F_{body}'$, that represents a plausible variation of the original.

This new function body is integrated back into the parent's base code to create a new program candidate, $X_{new}'$. This candidate is then executed and evaluated using the task-specific evaluator $T_t.E(\cdot)$, yielding a new score, $S_{new}'$. The new program, its function, and its score are registered as a new member of the low-level population $\mathcal{P}_{L,t}$. After the loop terminates, the procedure identifies the best-performing program in $\mathcal{P}_{L,t}$ and returns its optimized code, $X_t^*$, and final score, $S_t^*$.

## B.3  KNOWLEDGE TRANSFER

The `KnowledgeTransfer` procedure (Algorithm 4) is designed to enhance the multi-task proficiency of a newly generated individual, $I_{new}$, before it is integrated into the main population. This is achieved by systematically attempting to adapt its successful solutions from one task to another, leveraging the inherent relationships between tasks. The procedure takes the new individual $I_{new}$, the set of all tasks $\mathcal{T}$, and the LLM $\mathcal{L}$ as input.

The process operates through a series of pairwise comparisons across all tasks. It iterates through every possible source task, $t_{src}$, and target task, $t_{tgt}$, within the individual's repertoire. For each pair where $t_{src} \neq t_{tgt}$, the procedure attempts to transfer knowledge.

Specifically, it constructs a transfer-oriented prompt using `BuildTransferPrompt`. This prompt provides the LLM with the description of the target task ($T_{tgt}.D$), the full program code of the successful solution for the source task ($I_{new}.X_{new,t_{src}}$), and the code template for the target task ($T_{tgt}.Temp$). The LLM's objective is to synthesize this information and generate a new program, $X'_{transfer}$, that is a plausible adaptation of the source solution for the target context.

This newly generated program is immediately evaluated on the target task using its evaluator, $T_{tgt}.E(\cdot)$, to obtain a transfer score, $S'_{transfer}$. This score is then compared against the individual's existing score for the target task, $I_{new}.S_{new,t_{tgt}}$. If the transfer results in a performance improvement ($S'_{transfer} > I_{new}.S_{new,t_{tgt}}$), the individual is updated: its program and score for the target task, $t_{tgt}$, are replaced with the superior transferred versions, $X'_{transfer}$ and $S'_{transfer}$. After all possible transfers have been attempted, the potentially improved individual $I_{new}$ is returned.

---

### Prompt for Metaheuristic Generation

You are an expert algorithm designer. Your task is to create one novel algorithm for the following tasks:
{tasks_formatted}

Design and present the high-level task-agnostic pseudocode for your new algorithm refer to the following template.

```
ALGORITHM <Algorithm_Name>

/* PURPOSE: Brief description of the algorithm's purpose */

INPUT: <Description of input parameters/data>
OUTPUT: <Description of expected results/return values>

/* Initialization Phase */
Initialize necessary data structures, variables, or state
Set up initial conditions or constraints

/* Main Processing Loop (if applicable) */
WHILE termination criteria not satisfied DO
    Perform core algorithm operations
    Update algorithm state
    Evaluate progress or intermediate results
    Adjust parameters if needed
END WHILE

/* Post-Processing Phase (if applicable) */
Finalize results
Perform any cleanup or final transformations

RETURN output
```

- The pseudocode must describe the core strategy and logical flow of the algorithm at a conceptual level.
- Crucially, avoid low-level task-specific implementation details. Do not include specific variable names, data structures, or numerical constants.
- Ensure the pseudocode has a consistent shape ( 10–20 lines).

Enclose the entire pseudocode block within a single code block marked by '''pseudocode and '''.

---

---

### Prompt for Program Generation

You are an expert algorithm implementer. Given a pseudocode algorithm, convert it to an efficient Python implementation.

PSEUDOCODE:
{pseudocode}

IMPLEMENTATION REQUIREMENTS:
1. Use the template structure provided below
2. Ensure the implementation runs in acceptable time complexity
3. Maintain the core logic of the pseudocode
4. Use appropriate Python data structures and libraries

TEMPLATE:
{template_program_str}

RESPONSE FORMAT:
Return ONLY the Python code without explanations or examples, enclosed between "'python and "'
markers as shown:
"'python
# Your program here
"'

---

### Prompt for Key Function Identification

You are given a program. Please identify the most important function in this program that would
benefit most from optimization.
Program:
{program_str}

Task Description:
{task_description}

Return only the key function. It should be enclosed between "'python and"' markers exactly as shown
below:
"'python
# Your key function here
"'

---

### Prompt for Key Function Generation

You are given a function. Please create a variation of this function that with the same inputs and
outputs but might be more effective or use a different approach.
The function is part of a larger program solving the following task:

Task Description:
{task_description}

Original function body:
{original_function}

Return only the modified function. It should be enclosed between "'python and"' markers exactly as
shown below:
"'python
# Your key function here
"'

---

---

Prompt for Knowledge Transfer

You are an expert algorithm designer specialized in translating knowledge between related problems. Your task is to implement an algorithm for the specified target problem by drawing inspiration from a reference algorithm that solves a related task.

Target Problem:
{Task Description}

Reference Algorithm:
The following is a high-quality implementation for a related problem that can inform your approach:
{refer_program}

Implementation Template:
Your implementation should follow this template structure:
{template_program_str}

Return ONLY the Python code without explanations or examples, enclosed between "'python and "' markers as shown:
"'python
# Your program here
"'

---

**Algorithm 2** Initialization of High-Level Population

**Input:**
1: $\mathcal{T} = \{T_1, \ldots, T_m\}$: Set of tasks, each with description $D_t$
2: $\mathcal{L}$: Large Language Model
3: $N_H$: Target size for the high-level population
4: $N_L$: Low-level evaluation budget per individual
**Output:** Initialized high-level population $\mathcal{P}_H$
5: **procedure** INITIALIZEPOPULATION($\mathcal{T}, \mathcal{L}, N_H, N_L$)
6:     $\mathcal{P}_H \leftarrow \emptyset$
7:     prompt $\leftarrow$ BuildInitialPrompt($\{D_t\}_{t=1}^m$)
8:     InitialMHs $\leftarrow \mathcal{L}$(prompt)                    ▷ Generate a set of initial metaheuristics
9:     **for** each $MH_{init}$ in InitialMHs **do**
10:         **if** $N_H \geq N_H$ **then break**
11:         $I_{new} \leftarrow$ LowLevelEvolution($MH_{init}, \mathcal{T}, N_L, \mathcal{L}$)        ▷ Use main evaluation procedure
12:         **if** $I_{new}$ is valid **then**
13:             $\mathcal{P}_H \leftarrow \mathcal{P}_H \cup \{I_{new}\}$
14:     **return** $\mathcal{P}_H$

---

## C    PROBLEM AND EXPERIMENTAL SETTINGS

### C.1    TRAVELING SALESMAN PROBLEM (TSP)

**Problem Definition:**    Let $G = (V, E)$ be a complete graph where: $V = \{v_1, \ldots, v_n\}$ represents $n$ cities with coordinates $\mathbf{x}_i \in [0, 1]^2$ and $E$ contains edges with costs $c_{ij} = \|\mathbf{x}_i - \mathbf{x}_j\|_2$ (we consider Euclidean distance).

The objective is to find a Hamiltonian cycle $\pi = (\pi_1, \ldots, \pi_n, \pi_1)$ minimizing:

$$\mathcal{L}_{\text{TSP}} = \sum_{k=1}^{n-1} c_{\pi_k \pi_{k+1}} + c_{\pi_n \pi_1}.$$

**Task Description and Template:**    **TSP Task Description:** Develop an algorithm to address the Traveling Salesman Problem. The objective is to determine the shortest route that visits each city in a given list exactly once and then returns to the starting city, thereby minimizing the total distance traveled.

---

**Algorithm 3** Low-Level Key Function Evolution

---

**Input:**
1: $X_t$: Program code for task $t$
2: $F_t$: Identified key function for task $t$
3: $T_t$: Task object, containing description $D_t$ and evaluator $E_t(\cdot)$
4: $\mathcal{L}$: Large Language Model
5: $N_L$: Low-level evaluation budget (can be used to limit iterations)
**Output:** Optimized program code $X_t^*$ and its score $S_t^*$
6: **procedure** EVOLVEKEYFUNCTION($X_t, F_t, T_t, \mathcal{L}, N_L$)
7:     $\mathcal{P}_{L,t} \leftarrow$ Initialize with $(X_t, F_t, E_t(X_t))$         ▷ Seed low-level population
8:     $eval\_count\_L \leftarrow 1$
9:     **while** $eval\_count\_L < N_L$ **do**
10:         $p_{parent} \leftarrow \mathcal{P}_{L,t}$.Selection()     ▷ Select a program from the low-level pool
11:         prompt $\leftarrow$ BuildMutationPrompt($T_t.D, p_{parent}$.function)
12:         $F_{body}' \leftarrow \mathcal{L}$(prompt)         ▷ Mutate key function body
13:         $X_{new}' \leftarrow$ IntegrateFunction($p_{parent}$.code, $F_{body}'$)   ▷ Insert new function into base code
14:         $S_{new}' \leftarrow T_t.E(X_{new}')$
15:         $eval\_count\_L \leftarrow eval\_count\_L + 1$
16:         Register new program $(X_{new}', F_{body}', S_{new}')$ in $\mathcal{P}_{L,t}$
17:     $(X_t^*, S_t^*) \leftarrow$ GetBest($\mathcal{P}_{L,t}$)         ▷ Get code and score of the best program
18:     **return** $(X_t^*, S_t^*)$

---

**Algorithm 4** Knowledge Transfer

---

**Input:**
1: $I_{new}$: A new high-level individual with programs $\{X_{new,t}\}_{t=1}^m$ and scores $\{S_{new,t}\}_{t=1}^m$
2: $\mathcal{T} = \{T_1, \ldots, T_m\}$: Set of tasks
3: $\mathcal{L}$: Large Language Model
**Output:** Updated individual $I_{new}$
4: **procedure** KNOWLEDGETRANSFER($I_{new}, \mathcal{T}, \mathcal{L}$)
5:     **for** $t_{src} \leftarrow 1$ **to** $m$ **do**
6:         **for** $t_{tgt} \leftarrow 1$ **to** $m$ **do**
7:             **if** $t_{tgt} = t_{src}$ **then continue**
8:             prompt $\leftarrow$ BuildTransferPrompt($T_{tgt}.D, I_{new}.X_{new,t_{src}}, T_{tgt}.Temp$)
9:             $X_{transfer}' \leftarrow \mathcal{L}$(prompt)     ▷ Adapt source solution to target task
10:             $S_{transfer}' \leftarrow T_{tgt}.E(X_{transfer}')$
11:             **if** $S_{transfer}' < I_{new}.S_{new,t_{tgt}}$ **then**
12:                 $I_{new}.S_{new,t_{tgt}} \leftarrow S_{transfer}'$     ▷ Update score if transfer is successful
13:                 $I_{new}.X_{new,t_{tgt}} \leftarrow X_{transfer}'$     ▷ Update program code
14:     **return** $I_{new}$

---

**Training Instances:** For the heuristic evolution process, we use a set of 64 TSP instances, each with 100 locations randomly sampled from a uniform distribution over $[0, 1]^2$. The fitness of a candidate heuristic is measured by its average optimality gap, calculated against the optimal solutions found by the Concorde solver.

**Testing Instances:** For testing, we select commonly used 49 symmetric Euclidean TSPLib instances (Reinelt, 1991), with problem sizes ranging from 52 to 1,000 nodes.

## C.2 CAPACITATED VEHICLE ROUTING PROBLEM (CVRP)

**Problem Definition:** CVRP aims to minimize the total traveling distances of a fleet of vehicles given a depot and a set of customers with coordinates and demands. Given: 1) Depot $v_0$ and customers $\{v_1, ..., v_n\}$ with coordinates $\mathbf{x}_i \in [0, 1]^2$, 2) Demands $d_i \in \mathbb{Z}^+$ ($d_0 = 0$), 3) Vehicle capacity $Q \in \mathbb{Z}^+$, 4) Distance metric $c_{ij} = \|\mathbf{x}_i - \mathbf{x}_j\|_2$, find routes $\mathcal{R} = \{r_1, ..., r_m\}$. Each route $r_k$

**Template for TSP**

```python
import numpy as np

class TSPSolver:
    def __init__(self, coordinates: np.ndarray, distance_matrix: np.ndarray):
        """
        Initialize the TSP solver.

        Args:
            coordinates: Numpy array of shape (n, 2) containing the (x, y)
                coordinates of each city.
            distance_matrix: Numpy array of shape (n, n) containing pairwise
                distances between cities.
        """
        self.coordinates = coordinates
        self.distance_matrix = distance_matrix

        \# --- your code here ---

    def solve(self) -> np.ndarray:
        """
        Solve the Traveling Salesman Problem (TSP).

        Returns:
            A numpy array of shape (n,) containing a permutation of integers
            [0, 1, ..., n-1] representing the order in which the cities are
                visited.

            The tour must:
            - Start and end at the same city (implicitly, since it's a loop)
            - Visit each city exactly once
        """
        n = len(self.coordinates)

        \# --- your code here ---

        \# Example (naive ordered tour replace with your algorithm):
        tour = np.arange(n)

        return tour
```

starts/ends at $v_0$. Capacity constraints are satisfied $\sum_{v_i \in r_k} d_i \leq Q$ and all customers served exactly once. The objective is to minimize total distance.

**Task Description and Template:** **CVRP Task Description:** Develop an algorithm to solve the Capacitated Vehicle Routing Problem (CVRP). The objective is to determine the optimal set of routes for a fleet of vehicles that all start and end at a central depot. Each vehicle has a maximum capacity, and the routes must collectively serve all customer nodes exactly once without exceeding the vehicle's capacity. The goal is to minimize the total distance traveled across all routes.

**Training Instances:** The heuristic evolution is conducted on 64 randomly generated CVRP instances, each with 100 customers. Customer and depot locations are randomly sampled from $[0, 1]^2$. Each vehicle has a capacity of 50, and customer demands are integers sampled uniformly from $\{1, \ldots, 9\}$. The fitness value is the average gap to LKH solver (Helsgaun, 2017).

**Testing Instances:** We select 7 commonly used benchmark sets, including A, B, E, F, M, P, and X, from CVRPLib (Uchoa et al., 2017). The chosen sets and their characteristics are summarized in Table 4. Due to the time limit, we do not test on all instances from the X set.

Table 4: CVRPLib benchmark sets

| Benchmark Set | Number of Instances | Instance Size |
|---|---|---|
| Set A | 27 | 31-79 |
| Set B | 23 | 30-77 |
| Set E | 11 | 22-101 |
| Set F | 3 | 44-134 |
| Set M | 5 | 100-199 |
| Set P | 23 | 15-100 |
| Set X | 43 | 100-500 |

## C.3 FLOW SHOP SCHEDULING PROBLEM (FSSP)

**Problem Definition:** The Flow Shop Scheduling Problem (FSSP) aims to minimize the makespan (total time to complete all jobs) for a set of jobs that must be processed on a series of machines in a fixed order. Given: 1) A set of $n$ jobs $\mathcal{J} = \{J_1, ..., J_n\}$, 2) A set of $m$ machines $\mathcal{M} = \{M_1, ..., M_m\}$, 3) The processing time $p_{ij} \in \mathbb{Z}^+$ for each job $J_i$ on each machine $M_j$. The problem is to find a permutation (sequence) $\pi$ of the jobs. This sequence dictates the order in which jobs are processed on the first machine, and this same order is maintained for all subsequent machines. The objective is to find the sequence $\pi$ that minimizes the makespan, $C_{max}(\pi)$, which is the completion time of the last job on the last machine.

**Task Description and Template:** **FSSP Task Description:** Develop an algorithm to solve the Flow Shop Scheduling Problem (FSSP) by determining the optimal sequence of jobs to minimize makespan. In FSSP, all jobs must be processed on all machines in the same order (machine 0, then machine 1, then machine 2, etc.). The goal is to find the job sequence that minimizes the makespan (total completion time) while ensuring that: (1) all jobs follow the same machine processing order, (2) each machine processes only one job at a time, and (3) each job can only be processed on one machine at a time. The algorithm should return a permutation of job indices representing the order in which jobs should be processed.

**Training Instances:** For heuristic evolution, we use 64 randomly generated instances, each comprising 50 jobs and a number of machines varying between 2 and 20. The processing times for each job are sampled from a uniform distribution over $[0, 1]^2$. The average makespan (gap to lower bound) is used as the fitness value.

**Testing Instances:** We evaluate the algorithms on the widely-used Taillard instances (Taillard, 1993). We test 9 different test sets. The number of jobs in these instances ranges from 20 to 100, and the number of machines ranges from 5 to 20.

## C.4 BIN PACKING PROBLEM (BPP)

**Problem Definition:** We consider one-dimensional bin packing problem. The primary goal is to pack a set of $n$ items, each with a specific size or weight $w_j \in \mathbb{Z}^+$, into the minimum number of identical bins, each having a uniform capacity $C \in \mathbb{Z}^+$. The core challenge is to find a partition of the items into a set of bins $B = \{B_1, \ldots, B_m\}$ such that the sum of item sizes in any single bin does not exceed the capacity $C$, and the total number of bins used, $m$, is minimized.

**Task Description and Template:** **BPP Task Description:** You are given a set of items, each with a specific weight, and a number of identical bins, each with a fixed capacity. The goal is to pack all items into the minimum number of bins possible, such that the sum of the weights of the items in each bin does not exceed the bin's capacity.

**Instances:** Following the setup of Ye et al. (2024) and Zheng et al. (2025), we generate instances where bins have a capacity of 150 and item sizes are uniformly sampled from the range [20, 100].

### C.5 BLACK-BOX OPTIMIZATION (BBO)

**Problem Definition:** Black-box optimization (BBO) addresses the challenge of finding the minimum of an objective function $f(x)$ where its analytical form is unknown. The function $f : \mathbb{R}^d \to \mathbb{R}$ can only be evaluated at specific points $x$ to get its value, but its derivatives are unavailable. The goal is to find a vector $x^*$ within a bounded domain, $x \in [L, U]^d$, that minimizes the function's output, i.e., $x^* = \arg \min_x f(x)$. This problem is fundamental in many scientific and engineering fields, such as hyperparameter tuning and experimental design, where the relationship between inputs and outcomes is complex and can only be observed through evaluation.

**Task Description and Template:** **BBO Task Description:** You are tasked with implementing a general-purpose solver for black-box optimization problems. The solver must find a solution vector $x$ that minimizes a given objective function within a specified multi-dimensional search space. The solver will be initialized with the objective function, its dimensionality, and the search bounds. Your goal is to find the vector that results in the lowest possible function value.

**Instances:** The evaluation is performed on a set of five well-known benchmark functions for continuous optimization. Each function is tested in a 20-dimensional space ($d = 20$). The functions include unimodal (Sphere, Rosenbrock) and multimodal (Rastrigin, Ackley, Griewank) problems, providing a comprehensive test of the solver's ability to handle different optimization landscapes. Each function has a known global minimum of 0.0, and the search domains are defined as follows: Sphere [-10, 10], Rosenbrock [-5, 10], Rastrigin [-5.12, 5.12], Ackley [-32.768, 32.768], and Griewank [-600, 600].

### C.6 ADMISSIBLE SET PROBLEM (ASP)

**Problem Definition:** The Admissible Set Problem, rooted in extremal combinatorics, seeks to find the largest possible set of vectors (an "admissible set") that satisfies specific constraints. We focus on constructing a symmetric constant-weight admissible set, denoted as $I(n, w)$. This involves finding a set of vectors in $\{0, 1, \ldots, 6\}^k$ (where $n = 3k$) such that for any three distinct vectors $u, v, z$ from the set, there exists at least one coordinate position $i$ where the triplet $(u_i, v_i, z_i)$ is not a "bad triple". A "bad triple" is a predefined combination of values that is disallowed. The objective is to maximize the size of this admissible set. The problem has applications in areas like coding theory and the design of experiments.

**Task Description and Template:** **ASP Task Description:** You are given a vector, its dimension $n$, and its weight $w$. Your task is to assign a score to this vector. The score should reflect the vector's potential to be part of a large, valid admissible set. A higher score suggests that including this vector is more likely to lead to a larger final set. This scoring function will be used within a greedy algorithm to iteratively build the admissible set.

**Instances:** We evaluate the performance on standard benchmarks for this problem, defined by the dimension $n$ and weight $w$ of the vectors. The specific instances used are $(n = 15, w = 10)$, $(n = 12, w = 7)$, $(n = 21, w = 15)$, and $(n = 24, w = 17)$. The quality of the solution is measured by the size of the generated admissible set, with the goal of matching or exceeding known optimal sizes for these instances.

## D  MORE DETAILS ON METAHEURISTICS

Table 5 details the configuration and average running times of several metaheuristic solvers on standard TSPLib and CVRPLib benchmarks. We compare our Python implementations of Tabu Search, ALNS, and Memetic Search—accelerated using Numba, against Google's C++ OR-Tools solvers, an existing LLM-based AHD approach, and our proposed MTHS.

Table 5: Average Running Time and Configuration of Metaheuristic Solvers on Benchmark Datasets. Our Python-based implementations are compared against the highly optimized C++ solvers from Google OR-Tools, existing LLM-driven AHD approach and our proposed MTHS. Average running times are reported for standard TSPLib and CVRPLib instances.

| Category | Metaheuristic | Key Parameters | Key Accelerated Functions (Numba/C++) | Average Running Time | |
|---|---|---|---|---|---|
| | | | | TSPLib | CVRPLib |
| Our Python Implementations | Tabu Search (TS) | max_iterations: 100
tabu_tenure: 20 | _calculate_tour_distance_numba
_find_best_neighbor_numba | 58s | 8s |
| | Adaptive Large Neighborhood Search (ALNS) | max_iterations: 1000
removal_rate: [0.1, 0.4]
reaction_factor: 0.5 | _calculate_tour_cost
_greedy_insertion
_shaw_removal | 155s | 138s |
| | Memetic Search (MS) | population_size: 30
generations: 50
tournament_size: 5
patience: 40 | _calculate_tour_distance
_two_opt_local_search
_generate_nearest_neighbor_tour | 190s | 156s |
| OR-Tools Solvers | Tabu Search (TS)
Simulated Annealing (SA)
Guided Local Search (GLS) | Default
Default
Default | C++
C++
C++ | 60s
60s
60s | 60s
60s
60s |
| Existing AHD Approach | GLS
*(EoH, ReEvo, MCTS-AHD)* | iter_limit: 100
perturbation_moves: 30 | _two_opt_once
_relocate_once | 60–100s | 25–30s |
| MTHS (Ours) | ACSS
*(MTHS)* | time_limit: 100
population_size: 10 | _two_opt
_insert
_swap | 100s | 60s |

## E  MORE EXPERIMENTAL RESULTS AND ANALYSES

### E.1  MORE RESULTS ON TSP AND CVRP

To further assess the scalability and effectiveness of our approach, we test MTHS on larger TSPLib instances, with sizes ranging from 1000 to 2000 nodes. Table 6 presents the relative gap to the known optimal solutions for each instance. MTHS consistently finds high-quality solutions, often outperforming the other methods. For example, on the fl1577 and d1655 instances, MTHS achieves the smallest gaps of 1.23% and 3.37%, respectively. While some methods like GLS variants perform competitively on specific instances, our MTHS algorithm demonstrates a more robust and consistently strong performance across this challenging set of large-scale instances.

We evaluate MTHS on the an additional CVRP XML benchmark. The experiments are conducted on 64 randomly selected instances from the XML benchmark suite. As shown in Table 7, we report the average solution cost and the percentage gap relative to the state-of-the-art HGS solver, which serves as our baseline. MTHS demonstrates superior performance compared to all other commonly used metaheuristics and LLM-driven AHD methods, achieving an average gap of only 1.98%.

### E.2  PARETO FRONT

Figure 4 illustrates the search trajectory and final population of our proposed method, MTHS, in the three-dimensional objective space for the multi-task AHD. The background points represent all candidate metaheuristics in the population throughout the evolutionary process, colored by their generation index from early (dark purple) to late (bright yellow). This visualization demonstrates the algorithm's progression, showing how it initially explores a broad region of the objective space before intensifying its search and converging towards the Pareto front. The final, non-dominated population is highlighted in red, showcasing a well-distributed set of high-quality trade-off solutions

Table 6: Results on TSPLib instances of size 1000-2000.

| Method | vm1084 | pcb1173 | d1291 | fl1400 | fl1577 | d1655 | vm1748 | rl1889 |
|---|---|---|---|---|---|---|---|---|
| Constructive NN | 0.2598 | 0.2353 | 0.1799 | 0.3401 | 0.2558 | 0.2055 | 0.2125 | 0.2658 |
| Constructive Insert | 0.1875 | 0.2883 | 0.2390 | 0.0853 | 0.2408 | 0.2528 | 0.1728 | 0.1657 |
| OR-Tools SA | 0.0460 | 0.1429 | 0.1043 | 0.1321 | 0.1200 | 0.1203 | 0.1156 | 0.0908 |
| OR-Tools TS | 0.0448 | 0.1469 | 0.0859 | 0.1307 | 0.1225 | 0.1192 | 0.1011 | 0.0908 |
| ALNS | 0.0749 | 0.0833 | 0.1044 | 0.0437 | 0.0949 | 0.1186 | 0.0988 | 0.1073 |
| Tabu Search | 34.6348 | 5.7940 | 1.9034 | 0.4510 | 1.7804 | 1.7800 | 2.1891 | 3.4125 |
| Memetic Search | 0.0918 | 0.0893 | 0.0737 | 0.0565 | 0.0968 | 0.0994 | 0.0880 | 0.0940 |
| GLS EoH | 0.0548 | 0.0361 | 0.0569 | 0.0767 | 0.0367 | 0.0641 | 0.0308 | 0.0502 |
| GLS ReEvo | 0.0496 | 0.0383 | 0.0567 | 0.0767 | 0.0363 | 0.0636 | 0.0282 | 0.0502 |
| GLS MCTS | 0.0548 | 0.0386 | 0.0569 | 0.0767 | 0.0363 | 0.0641 | 0.0294 | 0.0502 |
| MHTS | 0.0360 | 0.0463 | 0.0341 | 0.0350 | 0.0123 | 0.0337 | 0.0339 | 0.0501 |

Table 7: Average gap to SOTA solver HGS on 64 XML instances.

| Method | Average Cost | Avg Relative Gap (%) |
|---|---|---|
| HGS (PyVRP) | 17953.40 | 0.00 |
| Constructive NN | 22563.79 | 30.42 |
| Constructive Insert | 23121.25 | 30.59 |
| OR-Tools GLS | 18429.38 | 2.79 |
| OR-Tools SA | 18852.62 | 5.70 |
| OR-Tools TS | 18365.94 | 2.54 |
| ALNS | 18663.75 | 3.55 |
| Tabu Search | 18858.68 | 5.72 |
| GLS EoH | 18537.64 | 2.92 |
| GLS ReEvo | 18593.80 | 3.23 |
| GLS MCTS | 18598.92 | 3.26 |
| Memetic Search | 21105.75 | 15.80 |
| **MTHS** | **18299.67** | **1.98** |

across the three conflicting objectives: TSP, CVRP, and FSSP. The distinct separation and advancement of the final front from the historical samples underscore the effectiveness of our approach in achieving both convergence and diversity.

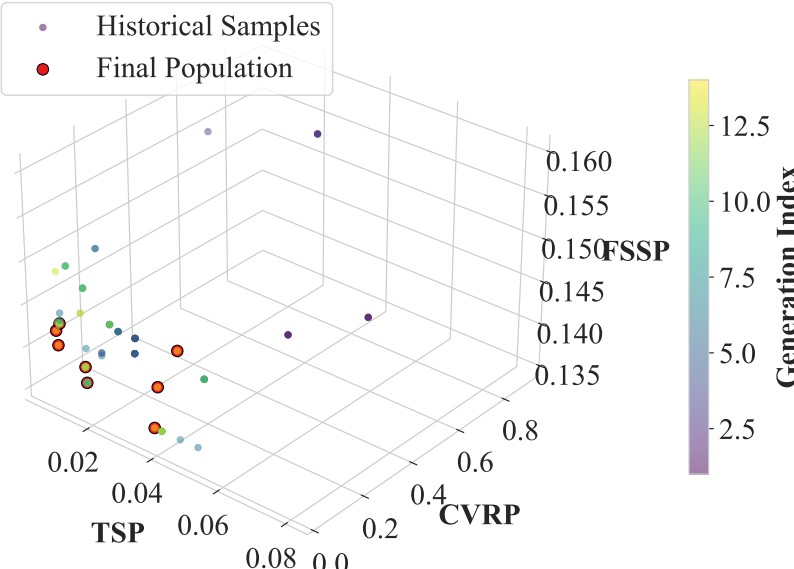

Figure 4: Metaheuristics generated by MTHS colored by generation and the final non-dominated front highlighted in red. The three objectives are fitness on three tasks (i.e., TSP, CVRP and FSSP). These metaheuristics that are removed during population management are not included.

### E.3 METAHEURISTIC REPRESENTATION

We identify and illustrate four distinct levels of abstraction for describing a metaheuristic algorithm: i) a high-level metaheuristic in MTHS, ii) an algorithmic pseudocode, iii) a code-level abstraction, and iv) a natural language thought description. The conceptual design outlines the overarching strategy, while pseudocode and code-level abstractions provide structured, implementation-oriented views. The thought description captures the core inventive idea in a dense, human-readable format.

For brevity and due to their structural similarity, we present a single example for the pseudocode and code-level abstraction formats. Each format is demonstrated below with a representative metaheuristic.

---

**Metaheuristic designed by MTHS**

**Adaptive Cooperative Substructure Search (ACSS)**

**Purpose:** A unified, task-agnostic metaheuristic to find high-quality feasible solutions for routing and scheduling problems by combining constructive heuristics, cooperative memory of useful substructures, adaptive perturbation, and constraint-aware repair.

> **INPUT:** A problem instance with a solution representation, an objective evaluator, and a constraint checker
> **OUTPUT:** A feasible solution (permutation or set of routes) with a near-optimal objective value

$\triangleright$ *Initialization Phase*

Construct a diverse set of initial candidate solutions using problem-aware constructive methods and randomization
Extract and record promising substructures from initial candidates into a cooperative memory

$\triangleright$ *Main Processing Loop*

**while** stopping condition not met **do**
    Select one or more candidates for improvement based on quality and diversity
    **Intensify:** apply local improvement operators guided by cooperative memory to reduce objective while preserving feasibility
    **Diversify:** apply adaptive, constraint-aware perturbations to escape local optima and generate varied neighborhood proposals
    **Repair:** enforce feasibility by applying generic constraint-handling procedures that adapt to problem specifics
    **Recombine:** optionally merge complementary substructures from cooperative memory into candidates to create new high-quality solutions
    Evaluate updated candidates with objective evaluator and constraint checker
    Update cooperative memory with newly discovered high-quality substructures and adjust operator selection probabilities based on recent success

$\triangleright$ *Post-Processing Phase*

Polish the best feasible solution using targeted local refinement and a final constraint-aware repair if needed

**RETURN** best feasible solution found

---

**Metaheuristic as Code Abstract/Pseudocode**

```
 1: procedure AMOCGS(problem, params)
 2:     population ← multi_construct(problem, params.heuristics)   ▷ task-specific constructive seeds
 3:     population ← map(lambda s: repair_and_evaluate(s, problem), population)         ▷ enforce
        constraints and score
 4:     operators ← init_operator_pool(problem)         ▷ problem-aware neighborhood & crossover
 5:     op_scores ← init_scores(operators); memory ← init_elite_memory(population)
 6:     best ← argmin(population)
 7:     while not termination_condition(params) do
 8:         parents ← select_parents(population, op_scores, params) ▷ biased by quality and diversity
 9:         op ← adaptive_select(operators, op_scores, params)
10:         offspring ← apply_operator(op, parents, problem)
11:         offspring ← local_search_and_repair(offspring, problem, params)        ▷ e.g., tabu/SA/LNS
        respecting constraints
12:         offspring.score ← evaluate(offspring, problem)
13:         population ← replace_population(population, offspring, params)        ▷ elite preservation +
        diversity maintenance
14:         op_scores ← update_op_scores(op_scores, op, offspring, improvement_metric(best, off-
        spring))
15:         best ← select_best(best, offspring)
16:         adapt_parameters(params, op_scores, population, memory)         ▷ temperature, operator
        weights, restart triggers
17:         if intensify_trigger(params) then
18:             path_relink_and_intensify(population, memory, problem)
19:     return best
```

> **Metaheuristic as Thought Description**
>
> Recursively partition the instance into manageable clusters (spatial for routing, temporal for schedul-ing), stochastically generate diverse candidate partial sequences within each cluster using lightweight local cost models, and iteratively merge clusters with a capacity- and precedence-aware repair operator that enforces feasibility; concurrently adapt sampling biases via online learning of high-value move patterns to concentrate search.
>
> During merges apply a multi-objective adaptive acceptance criterion that balances global cost reduction and constraint satisfaction, allowing focused local search and occasional exploratory perturbations to rapidly converge to high-quality feasible permutations and route sets.

## E.4 TOKENS AND COST

### E.4.1 COMPARISON OF DIFFERENT AHD METHODS

We conduct a detailed comparison of token consumption, number of evaluations, and wall-clock time against several baseline LLM-driven AHD methods. The results, averaged across the TSP and CVRP tasks, are presented below.

As shown in the table, our method (MTHS) achieves superior or comparable performance while being significantly more efficient. It requires fewer tokens and, critically, only one-third of the code evaluations compared to the baselines. This reduction in evaluations is a key advantage of our multi-task approach, leading to a substantial decrease in overall computational cost and making our framework more practical for real-world applications.

Table 8: Comparison of Token and Time Cost for AHD Methods.

| Method | Tokens (Approx.) | # Evaluations | Time Cost (Approx.) |
| --- | --- | --- | --- |
| *Traveling Salesperson Problem (TSP)* | | | |
| EoH | 3.0M | 1000 | 8 h |
| ReEvo | 3.2M | 1000 | 9 h |
| MCTS-AHD | 3.1M | 1000 | 40 h |
| **Ours (MTHS)** | **2.7M** | **333** | **8 h** |
| *Capacitated Vehicle Routing Problem (CVRP)* | | | |
| EoH | 3.2M | 1000 | 10 h |
| ReEvo | 3.3M | 1000 | 9 h |
| MCTS-AHD | 3.5M | 1000 | 45 h |
| **Ours (MTHS)** | **2.7M** | **333** | **8 h** |

### E.4.2 BREAKDOWN COST OF MTHS

We analyze the breakdown cost to generate and evaluate a single new metaheuristic individual, $I_{new}$, across $m$ tasks.

**High-Level Evolution**    This step generates one new metaheuristic, $MH_{new}$, from $k$ parents.

- **LLM Calls:** 1
- **Token Cost:** The input includes a prompt and $k$ parent metaheuristics; the output is $MH_{new}$.

**Low-Level Evolution**    This is the most expensive phase, executed for each of the $m$ tasks to eval-uate $MH_{new}$. The cost for a single task $T_t$ includes:

- **i) Program Generation:** 1 LLM call to combine $MH_{new}$ and a task template $Temp_t$ into a program $X_{new,t}$.

Table 9: Breakdown of token cost and number of LLM requests in MTHS.

| Type | Sub-type | Tokens/Sample | No. | Total Tokens | Percentage |
|---|---|---|---|---|---|
| High-level Evolution | Metaheuristic generation | 1k | 80 | 80k | 1% |
| Lower-level Evolution | Initial program generation | 6k | 240 | 1440k | 18% |
| | Key function identification | 4k | 240 | 960k | 12% |
| | Key function generation | 0.6k | 720 | 432k | 5% |
| | Program generation using new key function | 6k | 720 | 4320k | 52% |
| Knowledge Transfer | Program generation with knowledge transfer | 6k | 160 | 960k | 12% |
| **Total** | | | | **8192k** | **100%** |

- **ii) Key Function Identification:** 1 LLM call to analyze $X_{new,t}$ and extract the key function $F_{new,t}$.
- **iii) Key Function Refinement:** An evolutionary loop with a budget of $N_L$ evaluations, where each step uses the LLM as a mutation operator. This requires $N_L$ LLM calls.

The total cost for this stage scales linearly with the number of tasks ($m$) and the refinement budget ($N_L$).

**Knowledge Transfer**    After evaluation, this optional step adapts the best-performing program from a source task, $X^*_{new,src}$, to the other $m - 1$ target tasks.

- **LLM Calls:** $m - 1$
- **Token Cost:** Each call prompts the LLM with $X^*_{new,src}$ and a target task template.

**Summary of Costs**    The total number of LLM calls required to evaluate one new metaheuristic individual is:

$$Calls_{total} = \underbrace{1}_{\text{High-Level}} + \underbrace{m \times (2 + N_L)}_{\text{Low-Level}} + \underbrace{(m - 1)}_{\text{Knowledge Transfer}} \quad (1)$$

The dominant cost factor is the Low-Level Evolution, particularly the Key Function Refinement loop ($m \times N_L$ calls), making it the primary bottleneck in terms of time and expense.

When compared to existing LLM-driven AHD methods that target a single task, evaluating one MTHS individual requires a larger number of LLM requests due to the per-task evaluations. However, because MTHS simultaneously designs heuristics for multiple tasks within a single evolutionary run, the total computational budget required to find effective heuristics for an entire set of tasks is lower than running a single-task AHD method independently for each task. Table 9 lists the tokens used for each components in one run of MTHS on three tasks. It costs around 10 dollars when using GPT-5-mini.

### E.5    DETAILED RESULTS ON BENCHMARK INSTANCES

### E.6    ABLATION OF KEY COMPONENTS

Table 10: Detailed results for selected TSPLib instances (first seven instances in alphabetical order with different sizes and distributions): Gap Performance and Runtimes.

| Method | a280 | | berlin52 | | bier127 | | ch130 | | ch150 | | d198 | | d493 | |
|--------|------|------|----------|------|---------|------|-------|------|-------|------|------|------|------|------|
| | Gap | Time | Gap | Time | Gap | Time | Gap | Time | Gap | Time | Gap | Time | Gap | Time |
| NN | 27.43 | 0.23 | 19.08 | 0.01 | 14.86 | 0.19 | 23.98 | 0.19 | 25.53 | 0.21 | 19.33 | 0.21 | 24.02 | 0.34 |
| Insert | 20.13 | 0.02 | 4.55 | 0.00 | 12.02 | 0.00 | 6.31 | 0.00 | 9.45 | 0.01 | 12.48 | 0.01 | 24.48 | 0.06 |
| Or-tools SA | 4.39 | 60.13 | 4.80 | 60.04 | 2.25 | 60.18 | 1.73 | 60.39 | 1.67 | 60.11 | 1.23 | 60.01 | 3.29 | 60.68 |
| Or-tools TS | 4.67 | 60.23 | 0.03 | 60.04 | 1.43 | 60.18 | 1.73 | 60.42 | 1.67 | 60.14 | 1.23 | 60.25 | 3.58 | 60.17 |
| Or-tools GLS | 5.27 | 60.07 | 0.03 | 60.06 | 1.43 | 60.05 | 0.49 | 60.13 | 0.73 | 60.19 | 2.86 | 60.22 | 4.65 | 60.22 |
| MS | 6.24 | 35.30 | 0.03 | 1.98 | 1.03 | 7.08 | 3.53 | 6.59 | 2.85 | 3.64 | 2.24 | 13.78 | 4.76 | 409.47 |
| ALNS | 7.04 | 40.82 | 0.03 | 1.03 | 2.14 | 10.20 | 1.57 | 9.44 | 3.24 | 9.19 | 1.85 | 19.06 | 6.32 | 300.42 |
| TS | 11.32 | 124.54 | 3.42 | 70.16 | 6.37 | 83.09 | 4.23 | 87.29 | 5.81 | 92.50 | 3.73 | 116.51 | 9.46 | 113.10 |
| ACO_EoH | 27.77 | 39.56 | 1.79 | 4.59 | N/A | | 10.39 | 22.45 | 3.97 | 31.06 | 12.03 | 23.46 | 18.52 | 862.68 |
| ACO_MCTS | 9.50 | 83.46 | 0.03 | 28.92 | 3.23 | 18.79 | 3.40 | 27.90 | 1.63 | 29.78 | 2.61 | 20.38 | 18.88 | 909.84 |
| GLS_EoH | 1.78 | 351.41 | 0.03 | 2.72 | 0.04 | 2.02 | 1.15 | 4.49 | 0.84 | 4.10 | 0.95 | 259.85 | 1.66 | 563.71 |
| GLS_ReEvo | 2.94 | 349.71 | 0.03 | 3.09 | 0.62 | 2.51 | 0.64 | 5.62 | 0.97 | 4.39 | 1.16 | 400.06 | 2.72 | 563.09 |
| GLS_MCTS | 3.22 | 340.82 | 0.03 | 1.43 | 0.59 | 2.71 | 0.32 | 4.18 | 0.97 | 4.21 | 1.06 | 265.00 | 1.78 | 559.99 |
| STHS | 2.07 | 49.46 | 0.03 | 37.28 | 0.39 | 42.05 | 1.05 | 41.56 | 0.45 | 38.89 | 0.59 | 43.50 | 2.16 | 57.56 |
| MTHS | 1.34 | 100.17 | 0.03 | 81.26 | 0.39 | 100.00 | 0.64 | 100.00 | 0.37 | 100.01 | 0.39 | 100.00 | 1.63 | 100.01 |

### E.7 LLM TYPES

We evaluated our framework using four representative LLMs: two powerful commercial models (GPT-5-mini, Gemini-2.5-pro) and two leading open-source models (Deepseek-V3, Qwen3). The table below presents the performance (gap to best-known, lower is better) of one of the best meta-heuristics discovered by each LLM on three combinatorial optimization tasks. Our results indicate that while more capable models like GPT-5-mini and Gemini-2.5-pro tend to yield better overall performance, our method is robust and effective even when using open-source models. However, due to the complexity of metaheuristics, the more powerful LLMs usually generate better results.

### E.8 GENERALIZATION ON NEW PROBLEMS

We also investigated whether a metaheuristic designed by our framework can enhance the problem-solving capabilities of various LLMs on a new, unseen task (continuous black-box optimization). We prompted four different LLMs (GPT-5-mini, Gemini-2.5-pro, Claude-3.7, and GPT-3.5-turbo) to solve the task, both with and without the guidance of the metaheuristic (denoted as '+ MH'). The following tables report the average of the top-10 algorithms among 100 samples and the best score (lower is better). Results show that:

- The metaheuristic, originally designed for combinatorial optimization, generalizes effectively to the black-box optimization task despite its different structure and settings. We observe notable performance improvements across all four LLMs, regardless of their size and capabilities.

- Conversely, the metaheuristic does not generalize well to the admissible set task, where it generally provides no improvement. For instance, with GPT-5-mini, both the average and best scores worsened when using the metaheuristic, while only a slight improvement was observed for Gemini-2.5-pro.

Table 11: Detailed results for selected CVRPLib instances (middle-size X instances): Gap Performance and Runtimes.

| Instance | OR-Tools TS | | OR-Tools SA | | GLS EoH | | GLS ReEvo | | GLS MCTS | | MTHS | |
|---|---|---|---|---|---|---|---|---|---|---|---|---|
| | Gap | Time | Gap | Time | Gap | Time | Gap | Time | Gap | Time | Gap | Time |
| X-n303-k21 | 7.5 | 60.0 | 6.6 | 60.0 | 3.7 | 43.3 | 5.6 | 33.1 | 8.9 | 49.2 | 4.7 | 60.0 |
| X-n308-k13 | 8.9 | 60.0 | 10.4 | 60.0 | 5.4 | 28.9 | 5.9 | 29.7 | 8.4 | 47.2 | 6.8 | 60.0 |
| X-n313-k71 | 8.1 | 60.0 | 9.3 | 60.0 | 7.8 | 72.5 | 10.7 | 57.6 | 10.9 | 89.5 | 3.4 | 60.0 |
| X-n317-k53 | 1.3 | 60.0 | 1.3 | 60.0 | 1.5 | 39.9 | 1.4 | 40.5 | 1.4 | 70.2 | 1.4 | 60.0 |
| X-n322-k28 | 8.1 | 60.0 | 8.4 | 60.0 | 4.7 | 42.8 | 10.4 | 33.3 | 7.8 | 81.4 | 5.1 | 60.0 |
| X-n327-k20 | 7.6 | 60.0 | 7.4 | 60.0 | 3.4 | 38.3 | 5.9 | 31.8 | 6.5 | 51.7 | 6.2 | 60.0 |
| X-n331-k15 | 7.6 | 60.0 | 6.4 | 60.0 | 4.6 | 34.0 | 5.5 | 28.8 | 5.0 | 46.8 | 5.5 | 60.0 |
| X-n336-k84 | 4.0 | 60.0 | 4.1 | 60.0 | 4.8 | 93.0 | 4.4 | 71.1 | 4.8 | 95.2 | 3.8 | 60.0 |
| X-n344-k43 | 5.1 | 60.0 | 5.1 | 60.0 | 6.3 | 45.8 | 6.2 | 42.6 | 7.3 | 65.3 | 4.7 | 60.0 |
| X-n351-k40 | 9.7 | 60.0 | 9.1 | 60.0 | 6.0 | 65.6 | 8.4 | 52.6 | 9.2 | 73.8 | 4.4 | 60.0 |
| X-n359-k29 | 7.1 | 60.0 | 6.9 | 60.0 | 4.2 | 64.9 | 4.8 | 42.0 | 5.7 | 64.3 | 3.0 | 60.0 |
| X-n367-k17 | 10.0 | 60.0 | 6.8 | 60.0 | 10.0 | 86.5 | 9.5 | 108.6 | 8.6 | 182.0 | 10.6 | 60.0 |
| X-n376-k94 | 0.7 | 60.0 | 0.7 | 60.0 | 0.8 | 106.6 | 0.8 | 114.7 | 0.8 | 156.9 | 1.0 | 60.0 |
| X-n384-k52 | 5.6 | 60.0 | 5.3 | 60.0 | 4.9 | 135.4 | 5.7 | 101.4 | 5.0 | 160.9 | 3.8 | 60.0 |
| X-n393-k38 | 8.6 | 60.0 | 8.2 | 60.0 | 9.1 | 108.5 | 7.8 | 111.5 | 8.7 | 164.1 | 4.4 | 60.0 |
| X-n401-k29 | 3.7 | 60.0 | 3.7 | 60.0 | 3.2 | 155.4 | 5.3 | 162.6 | 3.7 | 233.3 | 2.5 | 60.0 |
| X-n411-k19 | 13.4 | 60.0 | 13.4 | 60.0 | 9.3 | 155.3 | 9.1 | 139.8 | 10.0 | 218.7 | 12.9 | 60.0 |
| X-n420-k130 | 6.4 | 60.0 | 6.9 | 60.2 | 4.9 | 180.1 | 4.7 | 159.5 | 5.3 | 221.2 | 5.0 | 60.0 |
| X-n429-k61 | 5.4 | 60.0 | 5.8 | 60.0 | 5.5 | 141.8 | 5.7 | 128.2 | 8.0 | 184.1 | 4.1 | 60.0 |
| X-n439-k37 | 4.7 | 60.0 | 4.9 | 60.1 | 3.0 | 140.2 | 2.9 | 120.5 | 3.3 | 180.6 | 5.2 | 60.0 |
| X-n449-k29 | 11.0 | 60.0 | 10.4 | 60.0 | 7.4 | 145.9 | 7.6 | 156.8 | 8.2 | 201.4 | 4.3 | 60.0 |
| X-n459-k26 | 12.6 | 60.0 | 10.6 | 60.0 | 12.8 | 161.8 | 10.4 | 168.9 | 10.4 | 207.4 | 7.9 | 60.0 |
| X-n469-k138 | 4.2 | 60.0 | 4.5 | 60.0 | 6.9 | 154.9 | 7.2 | 158.1 | 8.0 | 180.1 | 6.1 | 60.0 |
| X-n480-k70 | 4.1 | 60.0 | 4.0 | 60.0 | 4.7 | 139.2 | 5.3 | 137.5 | 5.6 | 169.4 | 3.7 | 60.0 |
| X-n491-k59 | 10.4 | 60.0 | 8.5 | 60.0 | 8.3 | 257.5 | 7.1 | 232.8 | 8.6 | 193.4 | 4.0 | 60.0 |

Table 12: Performance (gap to best-known) of the best metaheuristic found by different LLMs.

| LLM | TSP | CVRP | FSSP |
|---|---|---|---|
| GPT-5-mini | 0.0056 | 0.0412 | 0.1418 |
| Gemini-2.5-pro | 0.0083 | 0.0453 | 0.1399 |
| Qwen3 | 0.0244 | 0.0484 | 0.1454 |
| Deepseek-V3 | 0.0110 | 0.0556 | 0.1420 |

Table 13: Results on Black-box Optimization (lower is better).

| Method | Average Score | Best Score |
|---|---|---|
| GPT-5-mini | 2.8656 | 1.2192 |
| GPT-5-mini + MH | 0.3254 | 0.0265 |
| GPT-3.5-turbo | 21.7675 | 14.8752 |
| GPT-3.5-turbo + MH | 27.8859 | 3.7699 |
| Claude-3.7-sonnet | 0.5378 | 0.0000 |
| Claude-3.7-sonnet + MH | 0.4239 | 0.0000 |
| Gemini-2.5-pro | 10.8157 | 2.9874 |
| Gemini-2.5-pro + MH | 1.6639 | 0.3035 |

## E.9 KNOWLEDGE TRANSFER

Figure 7 shows an illustration of knowledge transfer from TSP to FSSP. We show the key structures for the three programs: the current best implementation for TSP, the implementation for FSSP before knowledge transfer, and the implementation for FSSP after knowledge transfer. The main referred parts, the original parts, and the revised parts after knowledge transfer are highlighted in blue, black, and red boxes, respectively. There are two knowledge transfer

- Change a sequential hybrid local search into two thorough search steps: best-insertion improvement and best-swap improvement.

- Transfer the implementation ideas on the main iterative search loop from TSP to FSSP: change a multi-start loop to a true iterative local search with repeated perturb and re-optimize steps

## F REPRODUCTION

We are committed to making our research fully reproducible and accessible to the broader community. We have made our code for metaheuristics and data publicly available. Our resources are hosted on an anonymous link `https://anonymous.4open.science/r/MTHS-E80B`.

The following components are provided:

1. **Detailed Experimental Results:** In the sections of this appendix, we present detailed tables and figures that elaborate on the results discussed in the main text. This includes per-instance performance and running times.

2. **Open-Sourced Algorithms:** The core contribution of our work, the generated metaheuristics, is available in our public repository. The code is commented to facilitate understanding and extension.

3. **Open-Sourced Evaluation Datasets and Scripts:** To ensure fair and consistent comparison, we have released the complete set of evaluation datasets, including TSP, CVRP, FSSP and BPP, used in our experiments. The repository also contains the exact scripts used to run the evaluations.

Table 14: Results on Admissible Set (lower is better).

| Method | Average Score | Best Score |
|---|---|---|
| GPT-5-mini | 631.5000 | 294.0000 |
| GPT-5-mini + MH | 899.1000 | 426.0000 |
| Gemini-2.5-pro | 824.1000 | 684.0000 |
| Gemini-2.5-pro + MH | 742.6500 | 279.0000 |

## G  USE OF LLMS

First, for manuscript preparation, the LLM was employed as a writing assistant to check grammar and refine phrasing, particularly in the introduction section. Second, the LLM was integrated as a core component of our proposed method to design and generate heuristics and programs.

**Template for CVRP**

```python
import numpy as np
class CVRPSolver:
    def __init__(self, coordinates: np.ndarray, distance_matrix: np.
        ndarray, demands: list, vehicle_capacity: int):
"""
Initialize the CVRP solver.

Args:
    coordinates: Numpy array of shape (n, 2) containing the (x, y)
        coordinates of each node, including the depot.
    distance_matrix: Numpy array of shape (n, n) containing pairwise
        distances between nodes.
    demands: List of integers representing the demand of each node (
        first node is typically the depot with zero demand).
    vehicle_capacity: Integer representing the maximum capacity of
        each vehicle.
"""
self.coordinates = coordinates
self.distance_matrix = distance_matrix
self.demands = demands
self.vehicle_capacity = vehicle_capacity

    \# --- your code here ---

    def solve(self) -> list:
"""
Solve the Capacitated Vehicle Routing Problem (CVRP).

Returns:
    A one-dimensional list of integers representing the sequence of
        nodes visited by all vehicles.
    The depot (node 0) is used to separate different vehicle routes
        and appears at the start and end
    of each route. For example: [0, 1, 4, 0, 2, 3, 0] represents:
 - Route 1: 0 - 1 - 4 - 0
 - Route 2: 0 - 2 - 3 - 0
"""
n = len(self.coordinates)

\# --- your code here ---

\# Example (naive solution replace with your algorithm):
solution = [0]  \# Start at the depot
current_capacity = 0

for i in range(1, n):
    if current_capacity + self.demands[i] > self.vehicle_capacity:
    solution.append(0)  \# return to depot and start a new route
    current_capacity = 0

    solution.append(i)
    current_capacity += self.demands[i]

if solution[-1] != 0:
    solution.append(0)  \# end the last route at the depot

return solution
```

**Template for FSSP**

```python
import numpy as np

class FSSPSolver:
    def __init__(self, num_jobs: int, num_machines: int,
            processing_times: list):
        """
        Initialize the FSSP solver.

        Args:
            num_jobs: Number of jobs in the problem
            num_machines: Number of machines in the problem
            processing_times: List of lists where processing_times[j][m] is
                the processing time of job j on machine m
        """
        self.num_jobs = num_jobs
        self.num_machines = num_machines
        self.processing_times = processing_times

        \# --- your code here ---

    def solve(self) -> list:
        """
        Solve the Flow Shop Scheduling Problem (FSSP).

        Returns:
            A list representing the sequence of jobs to be processed.
            For example, [0, 2, 1] means job 0 is processed first, then job
                2, then job 1.
            All jobs must be processed on all machines in the same order.

            The sequence must include all jobs exactly once.
        """

        \# --- your code here ---

        \# Simple solution: process jobs in their original order (0, 1, 2,
            ...)
        job_sequence = list(range(self.num_jobs))

        return job_sequence
```

**Template for BPP**

```python
import numpy as np

class BPPSolver:
    def __init__(self, capacity: int, weights: list[int | float]):
        """
        Initialize the BPP solver.

        Args:
            capacity (int): The capacity of each bin.
            weights (list[int | float]): A list of item weights.
        """
        self.capacity = capacity
        self.weights = weights
        self.num_items = len(weights)

        # --- your code here ---

    def solve(self) -> list[list[int]]:
        """
        Solve the Bin Packing Problem.

        Returns:
            A list of lists, where each inner list represents a bin and
                contains the
            original indices of the items packed into it.
            e.g., [[0, 2], [1, 3]] means item 0 and 2 are in the first bin,
             and item 1 and 3 are in the second.
        """

        # --- your code here ---

        bins = []  # Stores the content (indices) of each bin
        bin_loads = []  # Stores the current load of each bin

        # Store items as tuples of (index, weight) to keep track of original
            indices
        items = sorted([(i, w) for i, w in enumerate(self.weights)], key=
            lambda x: x[1], reverse=True)

        for item_index, item_weight in items:
            placed = False
            # Try to place the item in an existing bin
            for i in range(len(bins)):
                if bin_loads[i] + item_weight <= self.capacity:
                    bins[i].append(item_index)
                    bin_loads[i] += item_weight
                    placed = True
                    break

            # If not placed, open a new bin
            if not placed:
                bins.append([item_index])
                bin_loads.append(item_weight)

        return bins
```

**Template for BBO**

```python
import numpy as np
from typing import Callable, Tuple
class BBOSolver:
    def __init__(self,
                 objective_function: Callable[[np.ndarray], float],
                 dim: int,
                 bounds: Tuple[float, float]):
        """
        Initialize the Black-Box Optimization solver.
        Args:
            objective_function (Callable): The function to minimize.
                It takes a numpy
                                            array (vector) and returns
                                               a single float value.
            dim (int): The dimension of the input vector for the
                objective function.
            bounds (Tuple[float, float]): A tuple (min_val, max_val)
                representing the
                                          search space boundaries for
                                              each dimension.
        """
        self.objective_function = objective_function
        self.dim = dim
        self.bounds = bounds
        self.low, self.high = bounds

    def solve(self) -> np.ndarray:
        """
        Solve the optimization problem to find the minimum of the
            objective function.
        Returns:
            A numpy array representing the best solution vector found
                .
        """
        # --- Simple Random Search Implementation ---
        # This is a basic placeholder. You should implement a more
            sophisticated algorithm.
        num_iterations = 2000 * self.dim  # More iterations for
            higher dimensions
        best_solution = None
        best_value = float('inf')
        for _ in range(num_iterations):
            # Generate a random solution within the specified bounds
            current_solution = np.random.uniform(self.low, self.high,
                 self.dim)
            # Evaluate the solution
            current_value = self.objective_function(current_solution)
            # If this solution is better than the best one found so
                 far, update
            if current_value < best_value:
                best_value = current_value
                best_solution = current_solution
        # If no solution was found (e.g., num_iterations was 0),
            return a random one
        if best_solution is None:
            best_solution = np.random.uniform(self.low, self.high,
                self.dim)
        return best_solution
```

**Template for ASP**

```python
import math
import numpy as np

class ASSolver:
    def __init__(self):
        pass

    def solve(self, el: tuple[int, ...], n: int = 15, w: int = 10) ->
        float:
        """Returns the priority with which we want to add 'el' to the
            set.

        Args:
            el: A candidate vector. It's a tuple of integers.
            n: The dimension (length) of the vector 'el'.
            w: The weight of the vector 'el', a constraint on its
                elements.

        Returns:
            A float representing the priority score of the vector 'el
                '.
        """
        priority_score = 0.0
        return priority_score
```

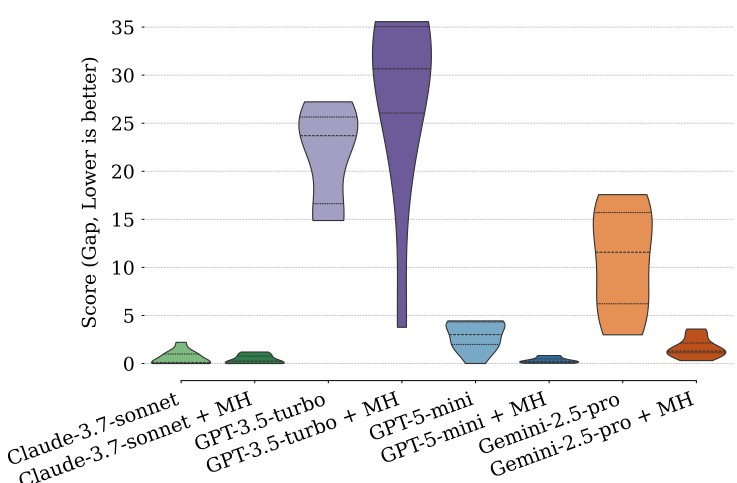

Figure 5: Generalization results on black-box optimization problem.

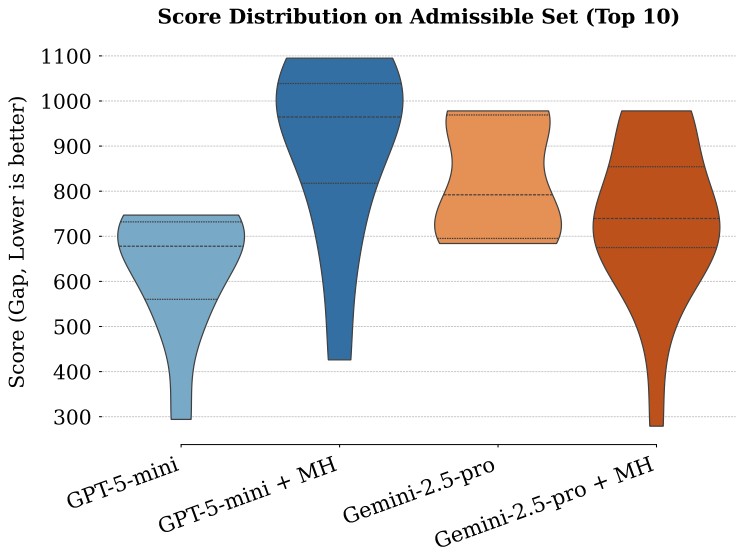

Figure 6: Generalization results on admissible set problem.

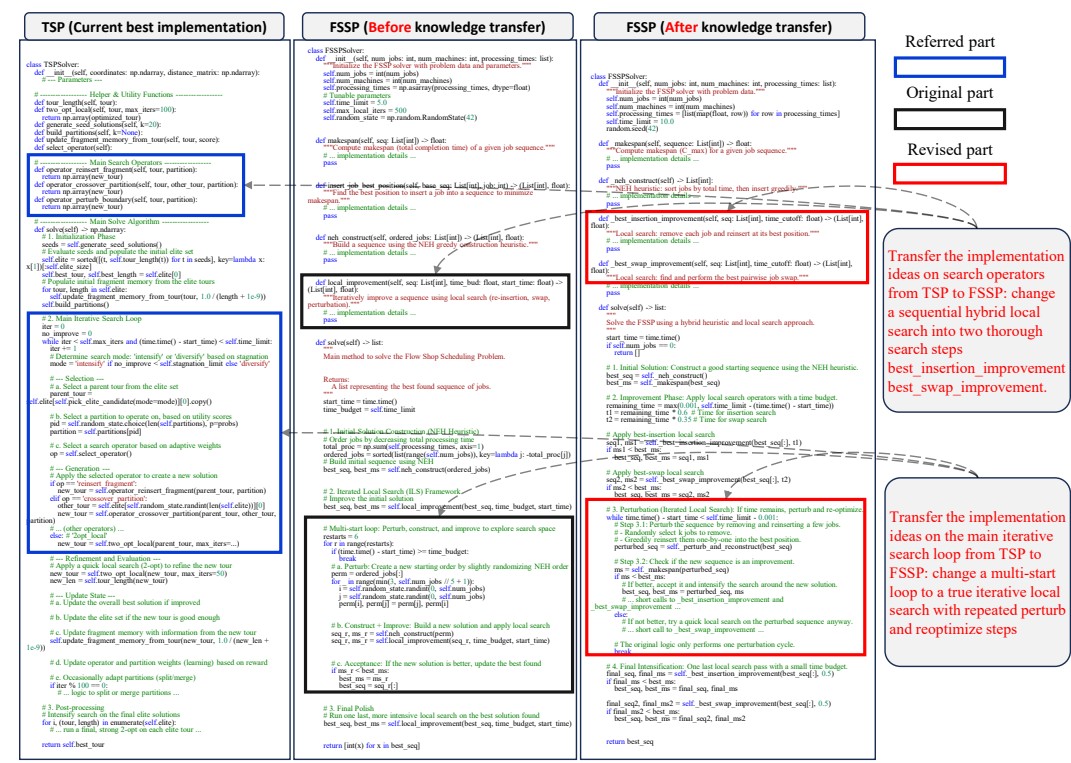

Figure 7: An illustration of knowledge transfer from TSP to FSSP. We show the key structures for the three progrmas: the current best implementation for TSP, the implementation for FSSP before knowledge transfer, and the implementation for FSSP after knowledge transfer. The main referred parts, the original parts and the revised parts after knowledge transfer are highlighted in blue, black and red boxes, respectively. We provide a brief summary of the two knowledge transfer points on the right side.

