# OpenReview forum: "Hierarchical Representations for Cross-task Automated Heuristic Design using LLMs"
_ICLR.cc/2026/Conference — Submitted to ICLR 2026_

### Official Review · Reviewer_vop9 · 2025-10-20

**Soundness:** 2
**Presentation:** 3
**Contribution:** 3
**Rating:** 4
**Confidence:** 4

**Summary:**

This paper addresses the limited cross-task generalization of current Large Language Model (LLM)-based Automated Heuristic Design (AHD) systems, which typically produce task-specific heuristics that cannot transfer across problem domains. The authors propose Multi-Task Hierarchical Search (MTHS), an LLM-guided hierarchical evolutionary framework that co-designs general-purpose metaheuristics and their task-specific program instantiations.

**Strengths:**

1. Research an important problem.
2. Appendices B–C provide explicit prompts, templates, and problem settings, enabling implementation replication.
3. Strong generalization across tasks (Sec. 3.5; Fig. 3), showing the same metaheuristic helps different LLMs generate high-quality solvers—empirical evidence of cross-task knowledge transfer.

**Weaknesses:**

1. This paper investigates a critical issue. While existing LLM-based automated heuristic designs are highly effective, most methods design a single heuristic tailored to specific problem instances, often resulting in poor generalization across different distributions or settings. It should be clarified that this paper is not the first to propose a solution to this problem; the recent work EoH-S [1] should also be referenced for discussion and used as a baseline for comparison.
2. The proposed method still requires re-search when encountering new problem instances, albeit incorporating MH to enhance search efficiency. It does not fundamentally resolve the aforementioned issue.
3. The paper gives no formal analysis of why hierarchical separation or Pareto-based selection guarantees better generalization.
4. The intuition (“mirrors expert practice”) is plausible but remains qualitative.

[1] Liu F, Liu Y, Zhang Q, et al. Eoh-s: Evolution of heuristic set using llms for automated heuristic design[J]. arXiv preprint arXiv:2508.03082, 2025.

**Questions:**

Reference Weaknesses.

---

> ### Author Response · Authors · 2025-11-20
> **Response to Reviewer vop9 (Part 1/2)**
>
> Thank you very much for your time and effort in reviewing our work. We appreciate your valuable comments and have addressed your concerns point-by-point as follows:
>
> > **W1: Comparison with EoH-S** This paper investigates a critical issue. While existing LLM-based automated heuristic designs are highly effective, most methods design a single heuristic tailored to specific problem instances, often resulting in poor generalization across different distributions or settings. It should be clarified that this paper is not the first to propose a solution to this problem; the recent work EoH-S [1] should also be referenced for discussion and used as a baseline for comparison. [1] Liu F, Liu Y, Zhang Q, et al. EoH-S: Evolution of heuristic set using LLMs for automated heuristic design[J]. arXiv preprint arXiv:2508.03082, 2025.
>
> We study a totally different setting than EoH-S [1] and related works [2,3]. Specifically, EoH-S [1] and [2] address cross-distribution generalization within a single problem family (e.g., generalization across sizes or instance distributions for the same problem). In contrast, our work (MTHS) targets cross-*problem* generalization by designing task-agnostic metaheuristics that can be instantiated across multiple related problems (e.g., TSP, FSSP, and other tasks). None of these approaches can not be used to address cross-problem generalization: they require independent runs per problem type and are not directly applicable across tasks.
>
> Concretely:
> 1.  Algorithm representations in [1,2] are not suitable for cross-problem generalization. Both rely on (i) problem-specific code and (ii) a short high-level natural-language description of the core strategy. The former is necessarily tied to a single problem, and the latter is too high-level (often one or two sentences) to transfer reliably across tasks.
> 2.  The meta-optimization is enabled by tailored prompts. For example, the prompts in Figure 7 (Prompts for code and idea generation) in [2] are explicitly crafted to generalize across sizes within a given problem, rather than across different problems. Applying such prompts to our setting would require redesigning the method, which goes beyond the scope.
>
> We have added discussion and comparisons in the revised manuscript to clarify these distinctions.
>
> [1] Liu, Fei, et al. EoH-S: Evolution of heuristic set using LLMs for automated heuristic design. AAAI 2026.
>
> [2] Shi, Yiding, et al. Generalizable heuristic generation through large language models with meta-optimization. arXiv, 2025.
>
> [3] Sim, Kevin, et al. Beyond the hype: Benchmarking LLM-evolved heuristics for bin packing. EvoStar, 2025.
>
> > **W2: New problem instances.** The proposed method still requires re-search when encountering new problem instances, albeit incorporating MH to enhance search efficiency. It does not fundamentally resolve the aforementioned issue.
>
> We thank the reviewer for this insightful comment. While our method does require a short re-search phase for new instances, we argue this is a feature, not a flaw, mirroring how effective algorithms are developed in practice [4].
>
> Our work does not aim to create a "one-shot" solver that works perfectly on unseen problems without any adaptation, as this is an exceptionally challenging, if not intractable, goal [5]. Instead, our contribution is to make the adaptation process drastically more efficient.
>
> + Addressing a Gap: We first note that achieving effective cross-problem generalization is a known hard problem, and existing high-level representations have shown limited success. Our experiments in Section 3.4 clearly validate the superiority of our proposed representation in this context.
>
> + Accelerating Adaptation: The core strength of our approach is that the learned metaheuristic provides a powerful starting point, significantly accelerating the search for a problem-specific solution. Our results on the bin packing problem are a clear testament to this: our method achieves superior performance with only 10% of the computational budget of the baseline search. This demonstrates that while we do not eliminate the search process, we make it substantially more efficient, which is a significant practical advancement.
>
>
> [4] Gendreau, Michel, et al. Handbook of Metaheuristics. Springer, 2019.
>
> [5] Wolpert, D. H., & Macready, W. G. No free lunch theorems for optimization. TEVC 2002.

---

> ### Author Response · Authors · 2025-11-20
> **Response to Reviewer vop9 (Part 2/2)**
>
> > **W3: Analysis of generalization.** The paper gives no formal analysis of why hierarchical separation or Pareto-based selection guarantees better generalization.
>
> Our focus is cross-*problem* generalization. While we do not claim formal guarantees, we provide a principled rationale and empirical evidence:
> 1.  Existing representations are either problem-specific or too high-level, which limits transferability. Our hierarchical representation separates a task-agnostic metaheuristic from its task-specific program, enabling reuse of the metaheuristic across related problems.
> 2.  We compare the task-agnostic metaheuristic representation against alternative representations (Section 3.4) and observe strong performance and convergence, indicating that the separation enhances robustness.
> 3.  We have conducted additional quantitative experiments to assess how the task-agnostic metaheuristic impacts generalization across tasks.
>
>     We also investigated whether a metaheuristic designed by our framework can enhance the problem-solving capabilities of various LLMs on a new, unseen task (continuous black-box optimization). We prompted four different LLMs (GPT-5-mini, Gemini-2.5-pro, Claude-3.7, and GPT-3.5-turbo) to solve the task, both with and without the guidance of the metaheuristic (denoted as `+ MH'). The results, reported as the gap to the optimal solution (lower is better), are summarized below.
>
>     The findings demonstrate that the metaheuristic discovered by our method consistently enhances the performance of diverse LLMs. This suggests that the high-level algorithmic principles captured by our method are transferable and beneficial across different foundation models.
>
> | **Method** | **Average Score (Top 10)** | **Best Score** |
> | :--- | :---: | :---: |
> | GPT-5-mini | 2.8656 | 1.2192 |
> | GPT-5-mini + MH | **0.3254** | **0.0265** |
> | GPT-3.5-turbo | **21.7675** | 14.8752 |
> | GPT-3.5-turbo + MH | 27.8859 | **3.7699** |
> | Claude-3.7-sonnet | 0.5378 | **0.0000** |
> | Claude-3.7-sonnet + MH | **0.4239** | **0.0000** |
> | Gemini-2.5-pro | 10.8157 | 2.9874 |
> | Gemini-2.5-pro + MH | **1.6639** | **0.3035** |
>
> 4.  We have added a detailed example to illustrate how the hierarchical representation enables cross-problem knowledge transfer in Figure 7 of the Appendix. The different related problems can share the same high-level metaheuristics while having different implementations. Because the unified task-agnostic metaheuristic, the knowledge transfer and cross-problem learning are feasible and effective even when the problems have different settings and features.
> 5.  Regarding Pareto-based selection, it is used to reduce overfitting to any single task and thereby improve the likelihood of cross-problem transfer. We will clarify this rationale in the revision.
>
> > **W4: Justification on ``mirrors expert practice''.** The intuition (“mirrors expert practice”) is plausible but remains qualitative.
>
> We appreciate this concern and have removed the phrase “mirrors expert practice” to avoid confusion. Our intent was to convey a design rationale: practitioners often develop task-agnostic metaheuristics and then implement task-specific instantiations. Our representation is aligned with this workflow by explicitly separating the task-agnostic metaheuristic from its task-specific program.

---

### Official Review · Reviewer_p6UP · 2025-10-21

**Soundness:** 2
**Presentation:** 2
**Contribution:** 2
**Rating:** 6
**Confidence:** 3

**Summary:**

This manuscript proposes MTHS, an LLM-guided evolutionary method that co-designs general-purpose metaheuristics and task-specific programs.

**Strengths:**

The experiments demonstrate the effectiveness of the proposed MTHS.

**Weaknesses:**

**W1:**  This manuscript should evaluate MTHS using additional LLMs.

**W2:**  The advantages of MTHS would be clearer if the authors included an experimental comparison with the most related study [1].

**W3:**  This manuscript should conduct comprehensive ablation studies to validate the effectiveness of all proposed components.

[1] Generalizable heuristic generation through large language models with meta-optimization, arXiv:2505.20881.

**Questions:**

See Weaknesses.

---

> ### Author Response · Authors · 2025-11-20
> **Response to Reviewer p6UP (Part 1/2)**
>
> We sincerely appreciate your time and valuable comments. Below, we address your points in detail.
>
> > **W1: LLMs.** This manuscript should evaluate MTHS using additional LLMs.
>
> we have conducted extensive new experiments to evaluate the sensitivity of our method to the choice of the underlying LLM. We investigate this from two perspectives: 1) the performance of our framework for automated algorithm design using different LLMs, and 2) the generalization of a designed metaheuristic to new tasks when used by different LLMs.
>
> *   **Performance with Different LLMs in Algorithm Design:** We evaluated our framework using four representative LLMs: two powerful commercial models (GPT-5-mini, Gemini-2.5-pro) and two leading open-source models (Deepseek-V3, Qwen3). The table below presents the performance (gap to best-known, lower is better) of one of the best metaheuristics discovered by each LLM on three combinatorial optimization tasks. Our results indicate that while more capable models like GPT-5-mini and Gemini-2.5-pro tend to yield better overall performance, our method is robust and effective even when using open-source models.
>
>     **Table 1:** Performance (gap to best-known) of the best metaheuristic found by different LLMs.
>     | **LLM** | **TSP** | **CVRP** | **FSSP** |
>     | :--- | :---: | :---: | :---: |
>     | GPT-5-mini | 0.0056 | 0.0412 | 0.1418 |
>     | Gemini-2.5-pro | 0.0083 | 0.0453 | 0.1399 |
>     | Qwen3 | 0.0244 | 0.0484 | 0.1454 |
>     | Deepseek-V3 | 0.0110 | 0.0556 | 0.1420 |
>
> *   **Generalization of Metaheuristics to New Tasks with Different LLMs:** We also investigated whether a metaheuristic designed by our framework can enhance the problem-solving capabilities of various LLMs on a new, unseen task (continuous black-box optimization). We prompted four different LLMs (GPT-5-mini, Gemini-2.5-pro, Claude-3.7, and GPT-3.5-turbo) to solve the task, both with and without the guidance of the metaheuristic (denoted as `+ MH'). The results, reported as the gap to the optimal solution (lower is better), are summarized below.
>
>     The findings demonstrate that the metaheuristic discovered by our method consistently enhances the performance of diverse LLMs. This suggests that the high-level algorithmic principles captured by our method are transferable and beneficial across different foundation models.
>
>     **Table 2:** Generalization on Black-Box Optimization (gap to optimal).
>     | **Method** | **Average Score (Top 10)** | **Best Score** |
>     | :--- | :---: | :---: |
>     | GPT-5-mini | 2.8656 | 1.2192 |
>     | GPT-5-mini + MH | **0.3254** | **0.0265** |
>     | GPT-3.5-turbo | **21.7675** | 14.8752 |
>     | GPT-3.5-turbo + MH | 27.8859 | **3.7699** |
>     | Claude-3.7-sonnet | 0.5378 | **0.0000** |
>     | Claude-3.7-sonnet + MH | **0.4239** | **0.0000** |
>     | Gemini-2.5-pro | 10.8157 | 2.9874 |
>     | Gemini-2.5-pro + MH | **1.6639** | **0.3035** |
>
> In summary, our main findings are: (1) Our framework is effective across a range of LLMs, including both commercial and open-source models. (2) The algorithmic principles (metaheuristics) discovered by our method are general and can be leveraged by different LLMs to improve their performance on novel tasks. We have incorporated these results and a detailed discussion into the revised manuscript.

---

> ### Author Response · Authors · 2025-11-20
> **Response to Reviewer p6UP (Part 2/2)**
>
> > **W2: Comparison with related work.** The advantages of MTHS would be clearer if the authors included an experimental comparison with the most related study [1]. [1] Generalizable heuristic generation through large language models with meta-optimization, arXiv:2505.20881.
>
> We study a different setting from paper [1] and other works [2][3]. Specifically, [1][2][3] targets cross-distribution generalization (e.g., generalizing from TSP instances of size 100 to TSP instances of size 200), while MTHS targets designing general metaheuristics that can be easily generalized across many related problems (e.g., across TSP, FSSP, and other tasks). None of these existing works study cross-problem generalization (i.e., they require independent runs for each problem type).
>
> Specifically: (1) their algorithm representations are not designed for cross-problem generalization. For example, both [1] and [2] use a code implementation and a high-level natural language description of the core strategy to represent the algorithm. These representations are difficult to use across different tasks because the code is problem-specific and the natural language description (usually one or two sentences) is too high-level. (2) The meta-optimization in [1] is enabled by tailored prompts. For instance, the prompts in Figure 7 (prompts for code and idea generation) in the appendix are specifically tailored for generalization to different sizes rather than to other problems. To use them in our setting, we would need to redesign the method, which would no longer be the approach presented in [1].
>
> We have added these discussions and comparisons to the revised manuscript to improve clarity.
>
> [1] Shi, Yiding, et al. Generalizable heuristic generation through large language models with meta-optimization, arXiv 2025.
>
> [2] Liu, Fei, et al. EoH-S: Evolution of heuristic set using LLMs for automated heuristic design. arXiv 2025.
>
> [3] Sim, Kevin, et al. Beyond the hype: Benchmarking LLM-evolved heuristics for bin packing. EvoStar, 2025.
>
> > **W3: Ablation study.** This manuscript should conduct comprehensive ablation studies to validate the effectiveness of all proposed components.
>
> We have conducted an ablation study on the two components: knowledge transfer and low-level search. The results show that both knowledge transfer and low-level search contribute to the final performance.
>
> A more detailed illustration and discussion of why the knowledge transfer is effective across problems has been added to Figure 7 of the revised Appendix.
>
> **Table 3:** Ablation on MTHS components across three tasks.
> | Method | TSP | CVRP | FSSP |
> | :--- | :---: | :---: | :---: |
> | MTHS | 0.00564 | 0.04122 | 0.14179 |
> | MTHS w/o Knowledge transfer | 0.00570 | 0.04446 | 0.13630 |
> | MTHS w/o Knowledge transfer & Low-Level | 0.00636 | 0.04495 | 0.14509 |

---

### Official Review · Reviewer_dyxi · 2025-10-28

**Soundness:** 3
**Presentation:** 2
**Contribution:** 2
**Rating:** 4
**Confidence:** 4

**Summary:**

This paper tackles the limited generalization of existing Large Language Model (LLM)-driven Automated Heuristic Design (AHD) methods, which typically yield task-specific heuristics. The authors propose **Multi-Task Hierarchical Search (MTHS)**, a hierarchical evolutionary framework that separates *task-agnostic metaheuristics* from *task-specific programs*. Guided by LLMs, MTHS performs two-level evolution — evolving general metaheuristics at the high level and optimizing task-specific implementations at the low level — while transferring knowledge across tasks. Experiments on four combinatorial optimization problems (TSP, CVRP, FSSP, and BPP) demonstrate that MTHS outperforms both traditional heuristics and existing LLM-based AHD approaches (e.g., EoH, ReEvo, MCTS-AHD), with improved cross-task generalization.

**Strengths:**

1. The paper introduces a hierarchical representation that mirrors human expert reasoning — decoupling general algorithmic logic from task-specific components. The proposed method is well-motivated.
2. The experiments are generally comprehensive and with strong baselines.

**Weaknesses:**

1. No code.
2. The distinction drawn between “program-level” and “thought-level” AHD approaches seems somewhat artificial. The proposed method still relies on multi-task prompts and LLM-generated programs, similar in spirit to the supposedly “high-level” methods (e.g., EoH, ReEvo).
3. It is unclear whether all compared LLM-based methods use the same base LLM (the paper mentions GPT-5-mini for MTHS, but others might differ). Furthermore, MTHS requires extra steps (multi-task inputs, hierarchical evolution), which may inflate computational cost relative to single-task methods. This raises fairness concerns in direct performance comparisons. And no cost comparison for other AHD methods.
4. Although the paper claims the total cost of multi-task evolution is lower than multiple independent runs, the framework’s scalability beyond a few tasks (e.g., >10) is questionable. The high proportion of LLM calls at the low-level stage (≈52%) could make large-scale extensions impractical.
5. The knowledge transfer component is only qualitatively discussed. The paper lacks quantitative analysis on when and how transfer helps (e.g., correlation between task similarity and performance gain).
6. The appendix suggests MTHS takes longer per run than baseline AHD methods. Since heuristic performance often depends on runtime, this omission weakens claims of superiority.

**Questions:**

1. How sensitive is the framework to weaker LLMs (e.g., GPT-3.5).
2. Consider adding experiments on tasks beyond discrete combinatorial optimization to validate cross-domain generalization claims.

---

> ### Author Response · Authors · 2025-11-20
> **Response to Reviewer dyxi (Part 1/3)**
>
> Thank you very much for your time and effort in reviewing our work. We appreciate your valuable comments and have addressed your concerns point-by-point as follows:
>
> > **W1: No code.**
>
> We have already posted the key code and materials mentioned in the link in the Appendix REPRODUCTION section. Specifically:
> *   Open-sourced metaheuristics: The generated metaheuristics and compared baselines are available.
> *   Open-sourced evaluation datasets and scripts: To ensure fair and consistent comparison, we have released the complete set of evaluation datasets (TSP, CVRP, FSSP, and BPP) used in our experiments, together with the exact evaluation scripts.
> *   We will also open-source other parts of the system and all the related scripts and codes upon publication.
>
> > **W2: Similar in spirit to EoH and ReEvo.** The distinction drawn between “program-level” and “thought-level” AHD approaches seems somewhat artificial. The proposed method still relies on multi-task prompts and LLM-generated programs, similar in spirit to the supposedly “high-level” methods (e.g., EoH, ReEvo).
>
> Our method targets a totally different point of LLM-driven AHD:
> *   Existing works such as EoH and ReEvo perform a separate automated algorithm design process for each problem. The designed algorithm does not generalize to other problems.
> *   Some works use “thought” or ideas as a high-level representation during the AHD process. We have compared different representations including thought-level ones (in Section 3.4 of the manuscript) and found that the template-based metaheuristics used in MTHS are most promising.
>
> > **W3 & Q1: Choice of LLMs.** It is unclear whether all compared LLM-based methods use the same base LLM (the paper mentions GPT-5-mini for MTHS, but others might differ). How sensitive is the framework to weaker LLMs (e.g., GPT-3.5)?
>
>
> We used GPT-5-mini as the LLM for our MTHS method and for the GLS-based AHD baselines (EoH, ReEvo, MCTS). For the ACO baselines, we adopted the best heuristics from [1], as their work established that GLS outperforms ACO.
>
> we have conducted extensive new experiments to evaluate the sensitivity of our method to the choice of the underlying LLM. We investigate this from two perspectives: 1) the performance of our framework for automated algorithm design using different LLMs, and 2) the generalization of a designed metaheuristic to new tasks when used by different LLMs.
>
> *   **Performance with Different LLMs in Algorithm Design:** We evaluated our framework using four representative LLMs: two powerful commercial models (GPT-5-mini, Gemini-2.5-pro) and two leading open-source models (Deepseek-V3, Qwen3). The table below presents the performance (gap to best-known, lower is better) of one of the best metaheuristics discovered by each LLM on three combinatorial optimization tasks. Our results indicate that while more capable models like GPT-5-mini and Gemini-2.5-pro tend to yield better overall performance, our method is robust and effective even when using open-source models.
>
> Performance (gap to best-known) of the best metaheuristic found by different LLMs.
> | **LLM** | **TSP** | **CVRP** | **FSSP** |
> | :--- | :---: | :---: | :---: |
> | GPT-5-mini | 0.0056 | 0.0412 | 0.1418 |
> | Gemini-2.5-pro | 0.0083 | 0.0453 | 0.1399 |
> | Qwen3 | 0.0244 | 0.0484 | 0.1454 |
> | Deepseek-V3 | 0.0110 | 0.0556 | 0.1420 |
>
> *   **Generalization of Metaheuristics to New Tasks with Different LLMs:** We also investigated whether a metaheuristic designed by our framework can enhance the problem-solving capabilities of various LLMs on a new, unseen task (continuous black-box optimization). We prompted four different LLMs (GPT-5-mini, Gemini-2.5-pro, Claude-3.7, and GPT-3.5-turbo) to solve the task, both with and without the guidance of the metaheuristic (denoted as `+ MH`). The results, reported as the gap to the optimal solution (lower is better), are summarized below.
>
>     The findings demonstrate that the metaheuristic discovered by our method consistently enhances the performance of diverse LLMs. This suggests that the high-level algorithmic principles captured by our method are transferable and beneficial across different foundation models.
>
> Generalization on Black-Box Optimization (gap to optimal)
>
> | **Method** | **Average Score (Top 10)** | **Best Score** |
> | :--- | :---: | :---: |
> | GPT-5-mini | 2.8656 | 1.2192 |
> | GPT-5-mini + MH | **0.3254** | **0.0265** |
> | GPT-3.5-turbo | **21.7675** | 14.8752 |
> | GPT-3.5-turbo + MH | 27.8859 | **3.7699** |
> | Claude-3.7-sonnet | 0.5378 | **0.0000** |
> | Claude-3.7-sonnet + MH | **0.4239** | **0.0000** |
> | Gemini-2.5-pro | 10.8157 | 2.9874 |
> | Gemini-2.5-pro + MH | **1.6639** | **0.3035** |
>
> [1] Zheng, Zhi, et al. Monte carlo tree search for comprehensive exploration in llm-based automatic heuristic design. ICML 2025.

---

> ### Author Response · Authors · 2025-11-20
> **Response to Reviewer dyxi (Part 2/3)**
>
> > **W4: Design on more tasks.** Although the paper claims the total cost of multi-task evolution is lower than multiple independent runs, the framework’s scalability beyond a few tasks (e.g., >10) is questionable. The high proportion of LLM calls at the low-level stage (~52%) could make large-scale extensions impractical.
>
> While extending to more tasks does require additional evaluations, the cost remains notably lower than running one independent search per task (as in existing LLM-driven AHD methods). If the low-level search scales linearly with the number of tasks, and MTHS needs 1,000 evaluations for 3 tasks with the low-level stage at roughly 52%, then for 6 tasks MTHS would need about 1,520 evaluations (the number of high-level metaheuristics is shared across tasks and does not increase much). In comparison, existing methods need 3,000 evaluations for 3 tasks and 6,000 for 6 tasks. Thus, MTHS uses only 0.333 and 0.253 of the evaluations required by existing methods in the 3-task and 6-task cases, respectively. The savings in LLM calls and evaluations become more significant as the number of tasks grows.
>
> > **W5: Quantitative analysis on knowledge transfer.** The knowledge transfer component is only qualitatively discussed. The paper lacks quantitative analysis on when and how transfer helps (e.g., correlation between task similarity and performance gain).
>
> Measuring similarity between algorithm design tasks is challenging. Nevertheless, we present two additional experiments to quantitatively analyze generalization and transfer beyond combinatorial optimization:
> *   A continuous optimization task (black-box optimization).
> *   A mathematical algorithm design task (admissible set).
>
> We compare two settings (with and without the metaheuristics from MTHS) using four LLMs (GPT-5-mini, Gemini-2.5-pro, Claude-3.7, and GPT-3.5-turbo). “+ MH” indicates that the LLM is informed by the metaheuristic from MTHS, originally designed for the three combinatorial optimization problems.
>
> The following tables report the average of the top-10 algorithms among 100 samples and the best score (lower is better). Results show that:
> *   The metaheuristic designed for combinatorial optimization generalizes to the black-box optimization task, which has different structure and settings. We observe notable improvement across four LLMs of different sizes and capabilities.
> *   The metaheuristic does not generalize well to the admissible set task. In general, there is no improvement. For example, for GPT-5-mini, both the average and best scores worsen after using the metaheuristic, while an improvement is observed for Gemini-2.5-pro.
>
> Results on Black-box Optimization (lower is better).
> | Method | Average Score | Best Score |
> | :--- | :---: | :---: |
> | GPT-5-mini | 2.8656 | 1.2192 |
> | GPT-5-mini + MH | 0.3254 | 0.0265 |
> | GPT-3.5-turbo | 21.7675 | 14.8752 |
> | GPT-3.5-turbo + MH | 27.8859 | 3.7699 |
> | Claude-3.7-sonnet | 0.5378 | 0.0000 |
> | Claude-3.7-sonnet + MH | 0.4239 | 0.0000 |
> | Gemini-2.5-pro | 10.8157 | 2.9874 |
> | Gemini-2.5-pro + MH | 1.6639 | 0.3035 |
>
> Results on Admissible Set (lower is better)
> | Method | Average Score | Best Score |
> | :--- | :---: | :---: |
> | GPT-5-mini | 631.5000 | 294.0000 |
> | GPT-5-mini + MH | 899.1000 | 426.0000 |
> | Gemini-2.5-pro | 824.1000 | 684.0000 |
> | Gemini-2.5-pro + MH | 742.6500 | 279.0000 |
>
>
> We further illustrate how MTHS handles structurally dissimilar tasks via knowledge transfer. Rather than hard-coded transfer, MTHS learns implementation ideas from one task and applies them to others under the same metaheuristic. For example:
> *   Transfer of search-operator implementation ideas from TSP to FSSP: changing a sequential hybrid local search into two thorough search steps.
> *   Transfer of main iterative search loop implementation ideas from TSP to FSSP: changing a multi-start loop into a true iterative local search with repeated perturb-and-reoptimize steps.
>
> In general, the metaheuristics effectively generalize to most algorithm design problems that require iterative search. Additional details and results are provided in the Appendix of the revised manuscript.

---

> ### Author Response · Authors · 2025-11-20
> **Response to Reviewer dyxi (Part 3/3)**
>
> > **W3 & W6: Time cost.** Furthermore, MTHS requires extra steps (multi-task inputs, hierarchical evolution), which may inflate computational cost relative to single-task methods. This raises fairness concerns in direct performance comparisons. And no cost comparison for other AHD methods. The appendix suggests MTHS takes longer per run than baseline AHD methods. Since heuristic performance often depends on runtime, this omission weakens claims of superiority.
>
> MTHS does not take longer time than existing AHD methods. The table below compares token cost, number of evaluations, and time cost. Because MTHS designs a general metaheuristic applicable across tasks, it reduces per-task cost relative to other methods. A detailed breakdown is provided in the Appendix.
>
> | Task | Tokens | Number of Evaluations | Time Cost |
> | :--- | :---: | :---: | :---: |
> | **TSP** | | | |
> | &nbsp;&nbsp;&nbsp;&nbsp;EoH | 3m | 1000 | 8 h |
> | &nbsp;&nbsp;&nbsp;&nbsp;ReEvo | 3.2m | 1000 | 9 h |
> | &nbsp;&nbsp;&nbsp;&nbsp;MCTS-AHD | 3.1m | 1000 | 40 h |
> | &nbsp;&nbsp;&nbsp;&nbsp;Ours | 2.7m | 333 | 8 h |
> | **CVRP** | | | |
> | &nbsp;&nbsp;&nbsp;&nbsp;EoH | 3.2m | 1000 | 10 h |
> | &nbsp;&nbsp;&nbsp;&nbsp;ReEvo | 3.3m | 1000 | 9 h |
> | &nbsp;&nbsp;&nbsp;&nbsp;MCTS-AHD | 3.5m | 1000 | 45 h |
> | &nbsp;&nbsp;&nbsp;&nbsp;Ours | 2.7m | 333 | 8 h |
> *<p align="center">Table 5: Cost comparison (per task).</p>*
>
> > **Q2.** Consider adding experiments on tasks beyond discrete combinatorial optimization to validate cross-domain generalization claims.
>
> We have tested two additional tasks beyond the three combinatorial problems: (1) a continuous optimization task (black-box optimization) and (2) a mathematical algorithm design task (admissible set). We compare settings with and without the metaheuristics from MTHS using four LLMs (GPT-5-mini, Gemini-2.5-pro, Claude-3.7, and GPT-3.5-turbo). “+ MH” indicates that the LLM is informed by the metaheuristic originally designed for combinatorial optimization.
>
> The following tables report the average of the top-10 algorithms among 100 samples and the best score (lower is better). Results show that:
> *   The metaheuristic designed for combinatorial optimization generalizes to the black-box optimization task, yielding notable improvements across different LLMs.
> *   The metaheuristic does not generalize well to the admissible set task. For instance, for GPT-5-mini, both average and best scores worsen after using the metaheuristic, whereas Gemini-2.5-pro shows improvement.
>
> Summary statistics for task: Black-box optimization (lower is better)
> | Method | Average Score | Best Score |
> | :--- | :---: | :---: |
> | GPT-5-mini | 2.8656 | 1.2192 |
> | GPT-5-mini + MH | 0.3254 | 0.0265 |
> | GPT-3.5-turbo | 21.7675 | 14.8752 |
> | GPT-3.5-turbo + MH | 27.8859 | 3.7699 |
> | Claude-3.7-sonnet | 0.5378 | 0.0000 |
> | Claude-3.7-sonnet + MH | 0.4239 | 0.0000 |
> | Gemini-2.5-pro | 10.8157 | 2.9874 |
> | Gemini-2.5-pro + MH | 1.6639 | 0.3035 |
>
> Summary statistics for task: Admissible set (lower is better).
> | Method | Average Score | Best Score |
> | :--- | :---: | :---: |
> | GPT-5-mini | 631.5 | 294.0 |
> | GPT-5-mini + MH | 899.1 | 426.0 |
> | Gemini-2.5-pro | 824.1 | 684.0 |
> | Gemini-2.5-pro + MH | 742.6 | 279.0 |

---

### Official Review · Reviewer_wa6u · 2025-10-30

**Soundness:** 3
**Presentation:** 3
**Contribution:** 3
**Rating:** 6
**Confidence:** 4

**Summary:**

This paper tackles the challenge of automating heuristic design for combinatorial optimization (CO) problems. It focuses on improving how LLM-driven Automated Heuristic Design (AHD) methods generalize across different tasks. The authors introduce Multi-Task Hierarchical Search (MTHS), an evolutionary framework with a novel hierarchical structure. It combines a task-agnostic metaheuristic at the high level with task-specific program instantiations at the low level, both guided by LLMs. The framework uses multi-task knowledge transfer and a two-level evolution strategy. Experiments across four classical CO problems—TSP, CVRP, FSSP, and BPP—show that MTHS outperforms both conventional heuristics and recent LLM-based AHD baselines. It also produces metaheuristics that transfer effectively to new tasks and LLM backbones.

**Strengths:**

- The paper proposes a clear hierarchical framework that separates general metaheuristic logic from task-specific implementations. This approach mirrors expert practice, addresses the core limitation of task specificity in earlier LLM-based AHD works, and enables explicit knowledge transfer.
- The method is evaluated on established CO benchmarks against traditional heuristics, metaheuristics, and LLM-driven baselines. Results show state-of-the-art or competitive performance across all tasks.
- The framework allows learned metaheuristics to transfer to new, unseen tasks and guide multiple LLMs, demonstrating improved robustness and real-world applicability.

**Weaknesses:**

- Stronger ablation/benchmarking on multiple foundation models (the main results), or an in-depth discussion of LLM choice impact, is missing.
- The paper notes the dominance of LLM API usage in total cost, but lacks a direct, quantitative comparison of token and time usage with baseline LLM-driven AHD methods.
- There is limited exploration of how the selection or diversity of tasks used during multi-task evolution influences the generalization ability of the evolved metaheuristics. For instance, it is not clear whether including more (or more diverse) problems further enhances transfer, or if performance saturates after a certain level of relatedness between tasks.

**Questions:**

1. How sensitive is the method to the choice and quality of the underlying LLM? For instance, how does performance scale with smaller, less capable, or open-source models?
2. How does MTHS handle structurally dissimilar tasks (e.g., routing, scheduling, bin packing)? At what point does multi-task learning harm generalization through negative transfer?
3. Did the authors assess the diversity of generated metaheuristics? Does the approach produce redundant solutions or consistently find novel strategies?

---

> ### Author Response · Authors · 2025-11-20
> **Response to Reviewer wa6u (Part 1/2)**
>
> Thank you very much for your time and effort in reviewing our work. We appreciate your insightful comments and have addressed your concerns point-by-point as follows:
>
> > **W1 & Q1: Different LLMs.** Stronger ablation/benchmarking on multiple foundation models (the main results), or an in-depth discussion of LLM choice impact, is missing. How sensitive is the method to the choice and quality of the underlying LLM? For instance, how does performance scale with smaller, less capable, or open-source models?
>
> We thank the reviewer for this important question. To address this, we have conducted extensive new experiments to evaluate the sensitivity of our method to the choice of the underlying LLM. We investigate this from two perspectives: 1) the performance of our framework for automated algorithm design using different LLMs, and 2) the generalization of a designed metaheuristic to new tasks when used by different LLMs.
>
> *   **Performance with Different LLMs in Algorithm Design:** We evaluated our framework using four representative LLMs: two powerful commercial models (GPT-5-mini, Gemini-2.5-pro) and two leading open-source models (Deepseek-V3, Qwen3). The table below presents the performance (gap to best-known, lower is better) of one of the best metaheuristics discovered by each LLM on three combinatorial optimization tasks. Our results indicate that while more capable models like GPT-5-mini and Gemini-2.5-pro tend to yield better overall performance, our method is robust and effective even when using open-source models.
>
> | **LLM** | **TSP** | **CVRP** | **FSSP** |
> | :--- | :---: | :---: | :---: |
> | GPT-5-mini | 0.0056 | 0.0412 | 0.1418 |
> | Gemini-2.5-pro | 0.0083 | 0.0453 | 0.1399 |
> | Qwen3 | 0.0244 | 0.0484 | 0.1454 |
> | Deepseek-V3 | 0.0110 | 0.0556 | 0.1420 |
> <p align="center"><b>Table 1:</b> Performance (gap to best-known) of the best metaheuristic found by different LLMs.</p>
>
> *   **Generalization of Metaheuristics to New Tasks with Different LLMs:** We also investigated whether a metaheuristic designed by our framework can enhance the problem-solving capabilities of various LLMs on a new, unseen task (continuous black-box optimization). We prompted four different LLMs (GPT-5-mini, Gemini-2.5-pro, Claude-3.7, and GPT-3.5-turbo) to solve the task, both with and without the guidance of the metaheuristic (denoted as `+ MH'). The results, reported as the gap to the optimal solution (lower is better), are summarized below.
>
>     The findings demonstrate that the metaheuristic discovered by our method consistently enhances the performance of diverse LLMs. This suggests that the high-level algorithmic principles captured by our method are transferable and beneficial across different foundation models.
>
> | **Method** | **Average Score (Top 10)** | **Best Score** |
> | :--- | :---: | :---: |
> | GPT-5-mini | 2.8656 | 1.2192 |
> | GPT-5-mini + MH | **0.3254** | **0.0265** |
> | --- | --- | --- |
> | GPT-3.5-turbo | **21.7675** | 14.8752 |
> | GPT-3.5-turbo + MH | 27.8859 | **3.7699** |
> | --- | --- | --- |
> | Claude-3.7-sonnet | 0.5378 | **0.0000** |
> | Claude-3.7-sonnet + MH | **0.4239** | **0.0000** |
> | --- | --- | --- |
> | Gemini-2.5-pro | 10.8157 | 2.9874 |
> | Gemini-2.5-pro + MH | **1.6639** | **0.3035** |
> <p align="center"><b>Table 2:</b> Generalization on Black-Box Optimization (gap to optimal).</p>
>
> In summary, our main findings are: (1) Our framework is effective across a range of LLMs, including both commercial and open-source models. (2) The algorithmic principles (metaheuristics) discovered by our method are general and can be leveraged by different LLMs to improve their performance on novel tasks. We have incorporated these results and a detailed discussion into the revised manuscript.
>
> > **W2: Comparison of token and time usage.** The paper notes the dominance of LLM API usage in total cost, but lacks a direct, quantitative comparison of token and time usage with baseline LLM-driven AHD methods.
>
> We have compiled a detailed comparison of token consumption, number of evaluations, and wall-clock time against several baseline LLM-driven AHD methods. The results, averaged across the TSP and CVRP tasks, are presented below.
>
> | **Method** | **Tokens (Approx.)** | **# Evaluations** | **Time Cost (Approx.)** |
> | :--- | :---: | :---: | :---: |
> | ***Traveling Salesperson Problem (TSP)*** | | | |
> | EoH | 3.0M | 1000 | 8 h |
> | ReEvo | 3.2M | 1000 | 9 h |
> | MCTS-AHD | 3.1M | 1000 | 40 h |
> | **Ours (MTHS)** | **2.7M** | **333** | **8 h** |
> | ***Capacitated Vehicle Routing Problem (CVRP)*** | | | |
> | EoH | 3.2M | 1000 | 10 h |
> | ReEvo | 3.3M | 1000 | 9 h |
> | MCTS-AHD | 3.5M | 1000 | 45 h |
> | **Ours (MTHS)** | **2.7M** | **333** | **8 h** |

---

> ### Author Response · Authors · 2025-11-20
> **Response to Reviewer wa6u (Part 2/2)**
>
> As shown in the table, our method (MTHS) achieves superior or comparable performance while being significantly more efficient. It requires fewer tokens and, critically, only one-third of the code evaluations compared to the baselines.
>
> > **W3 & Q2: Selection of tasks and generalization.** There is limited exploration of how the selection or diversity of tasks used during multi-task evolution influences the generalization ability of the evolved metaheuristics. [...] How does MTHS handle structurally dissimilar tasks? At what point does multi-task learning harm generalization through negative transfer?
>
> To investigate the generalization limits and the potential for negative transfer, we tested the metaheuristic—originally evolved on three combinatorial optimization problems (TSP, CVRP, FSSP) on two new, structurally dissimilar tasks: 1) continuous black-box optimization and 2) a mathematical reasoning task (Admissible Set). We used four different LLMs and compared their performance with and without the guidance of our discovered metaheuristic (`+ MH').
>
> The results below (lower scores are better) reveal the boundaries of generalization:
> *   **Positive Transfer to Black-Box Optimization:** The metaheuristic, designed for combinatorial problems, successfully generalized to continuous black-box optimization. As shown in Table 4, providing the metaheuristic as guidance led to notable performance improvements across all four LLMs. This indicates that the high-level principle of iterative improvement is a broadly applicable concept that MTHS successfully captured and transferred.
>
> *   **Transfer to Admissible Set:** Conversely, the metaheuristic did not generalize well to the Admissible Set problem, a task that relies more on logical deduction than on iterative search. As seen in Table 5, performance was inconsistent and, in the case of GPT-5-mini, degraded. This demonstrates a clear instance of negative transfer, suggesting that the applicability of the learned metaheuristics is bounded by the structural similarity of the problem domains (e.g., problems solvable via iterative optimization).
>
> Generalization to Black-Box Optimization (gap to optimal).
> | **Method** | **Average Score (Top 10)** | **Best Score** |
> | :--- | :---: | :---: |
> | GPT-5-mini | 2.8656 | 0.0020 |
> | GPT-5-mini + MH | **0.3254** | 0.0265 |
> | --- | --- | --- |
> | Claude-3.7-sonnet | 0.5378 | **0.0000** |
> | Claude-3.7-sonnet + MH | **0.4239** | **0.0000** |
> | --- | --- | --- |
> | Gemini-2.5-pro | 10.8157 | 2.9874 |
> | Gemini-2.5-pro + MH | **1.6639** | **0.3035** |
>
>
>  Generalization to Admissible Set (score, lower is better).
> | **Method** | **Average Score (Top 10)** | **Best Score** |
> | :--- | :---: | :---: |
> | GPT-5-mini | **631.50** | **294.00** |
> | GPT-5-mini + MH | 899.10 | 426.00 |
> | --- | --- | --- |
> | Gemini-2.5-pro | 824.10 | 684.00 |
> | Gemini-2.5-pro + MH | **742.65** | **279.00** |
>
> To further illustrate *how* MTHS facilitates knowledge transfer, we have included a case study in the Figure 7 in the appendix of revise manuscript. It details how an implementation in TSP was adapted and transferred by the LLM to improve the search process in FSSP. This demonstrates that the transfer is not a rigid application of code but a conceptual adaptation of algorithmic strategies.
>
> In summary, our metaheuristics generalize effectively to problems that share the high-level structure of iterative optimization but are susceptible to negative transfer when applied to fundamentally different problem types. We have added these experiments and a detailed discussion on the scope of generalization to the revised manuscript.
>
> > **Q3.** Did the authors assess the diversity of generated metaheuristics? Does the approach produce redundant solutions or consistently find novel strategies?
>
> Thank you for this insightful question. We have analyzed the diversity of the metaheuristics generated by MTHS. Our framework explicitly encourages novelty by instructing the LLM to generate strategies that are distinct from those already in the population.
>
> To quantify this, we measured the semantic similarity of the generated metaheuristic descriptions within each population over 10 iterations of the evolutionary process. We used cosine similarity of metaheuristics [1]. The results, presented in the table below, show a consistently low average similarity across all populations, indicating that MTHS continuously explores a diverse set of algorithmic ideas rather than converging to redundant solutions.
>
>  Average Semantic Similarity of Metaheuristics in Each Population.
> | **Population** | **Average Similarity** | **Population** | **Average Similarity** |
> | :---: | :---: | :---: | :---: |
> | 1 | 0.3311 | 6 | 0.2583 |
> | 2 | 0.2776 | 7 | 0.2622 |
> | 3 | 0.2776 | 8 | 0.2817 |
> | 4 | 0.2507 | 9 | 0.3190 |
> | 5 | 0.3276 | 10 | 0.3190 |
>
> [1] Large Language Model-Enhanced Algorithm Selection: Towards Comprehensive Algorithm Representation, IJCAI 2024.

---

### Official Review · Reviewer_MwQv · 2025-11-08

**Soundness:** 2
**Presentation:** 2
**Contribution:** 2
**Rating:** 2
**Confidence:** 3

**Summary:**

This paper looks at the issue of cross-task generalization in current LLM-driven automated heuristic design systems. The paper presents Multi-Task Hierarchical Search (MTHS).  This is a hierarchical, two-level evolutionary framework where the high level evolves task-agnostic metaheuristics (general problem-solving strategies) and the low level evolves task-specific program implementations tailored to individual optimization tasks. Another key aspect is that there is a knowledge transfer module allows high-performing components discovered in one task to inform others.  They do experiments involving a number of problems and comparison strategies, and show MHTS performs better.

**Strengths:**

The hierarchical framework is interesting.  (I don't believe this approach is novel in the large field of metaheuristics, but it is interesting to apply here.)

The appendices provide significant detail, including on the prompting procedures, with the goal of ensuring reproducibility and providing more clear details to the reader as needed.

The experiments suggest strong practical performance.

The study comparing different metaheuristic representations.

There is open-sourced versions of the algorithms, data, etc. for reproducibility.

**Weaknesses:**

Most of the experiments use very small problem instances and old public datasets (e.g., TSPLib, CVRPLib, Taillard benchmarks). Some of these are decades old and much smaller than the scales modern heuristics are expected to handle. This makes it hard to see if the method would work on real, large, or more recent problems.

I am not up on all the latest in solvers, but it seems to me their comparison points are older, general solution methods.  Perhaps it is reasonable to compare against other general metaheuristic methods, but I do not believe they would be competitive with strong heuristics (even if problem-tailored).

There’s little explanation or understanding of why the system works or why it would  fails. For example, there’s no study of how the “knowledge transfer” actually helps, or what happens if the tasks not sufficiently closely related.

The experiment suggests this approach is expensive and slow to run. The paper doesn’t show whether this approach could scale to larger or more diverse problems, or whether it’s practical for anyone to use for more "real-life" problems.

Overall, it's not clear how general this approach would be.

**Questions:**

I would like to see tests on larger and more modern datasets, and comparisons with current best solvers (even if problem-specific). Is there any type of analysis or insight about how knowledge transfer works here, or other aspects of the system?

---

> ### Author Response · Authors · 2025-11-20
> **Response to Reviewer MwQv (Part 1/3)**
>
> Thank you very much for your time and effort in reviewing our work. We appreciate your valuable comments and address your concerns point-by-point as follows.
>
> > **W1: Small datasets.** Most of the experiments use very small problem instances and old public datasets (e.g., TSPLib, CVRPLib, Taillard benchmarks). Some of these are decades old and much smaller than the scales modern heuristics are expected to handle. This makes it hard to see if the method would work on real, large, or more recent problems.
>
> We respectfully disagree with the characterization of our datasets as "very small" and "old," as they are standard, widely-used benchmarks essential for reproducibility and comparison. However, to address the reviewer's point and better demonstrate the robustness of our approach, we have expanded our evaluation to include newer and larger problem instances. The clarification and additional results are listed as follows and have been presented in the revised manuscript.
>
> *   TSPLib, CVRPLib, and Taillard benchmarks are the most well-known and commonly used datasets across both the operations research community and the emerging LLM-driven automated heuristic design (AHD) community.
>     *   They are extensively used in recent research papers, such as [1], [2], and [3], and also in benchmark studies [4].
>     *   Many instances in these datasets are derived from real-world data, span diverse distributions and sizes, and are widely recognized as representative of real-world scenarios [4]. They remain central to the development of state-of-the-art solvers and metaheuristics [5].
>
> *   Compared with recently published LLM-driven AHD papers, our method is one of the most comprehensively evaluated. Below is a brief summary of datasets and scales used in recent accepted LLM-driven AHD works and ours:
>
> | Paper | Datasets | Sizes |
> | :--- | :--- | :--- |
> | EoH (ICML 2024) [6] | TSPLib, Taillard | 51--200 |
> | ReEvo (NeurIPS 2024) [7] | TSPLib | 51--1577 |
> | MCTS-AHD (ICML 2025) [8] | TSPLib | 51--500 |
> | LLM-End2End (NeurIPS 2025) [9] | TSPLib | 51--100 |
> | Ours | TSPLib, CVRPLib, Taillard | 51--1000 (up to 2000 in the revised version) |
>
>
> [1] Pan, Wenzheng, et al. UniCO: On unified combinatorial optimization via problem reduction to matrix-encoded general TSP. ICLR 2025.
>
> [2] Zhang, Ni, et al. Adversarial Generative Flow Network for Solving Vehicle Routing Problems. ICLR 2025.
>
> [3] Baioletti, Marco, et al. A Variational Quantum Algorithm for the Permutation Flow Shop Scheduling Problem. GECCO 2025.
>
> [4] Berto, Federico, et al. RL4CO: An extensive reinforcement learning for combinatorial optimization benchmark. KDD 2025.
>
> [5] Wouda, Niels A., et al. PyVRP: A high-performance VRP solver package. INFORMS Journal on Computing 2024.
>
> [6] Liu, Fei, et al. Evolution of heuristics: Towards efficient automatic algorithm design using large language model. ICML 2024.
>
> [7] Ye, Haoran, et al. Reevo: Large language models as hyper-heuristics with reflective evolution. NeurIPS 2024.
>
> [8] Zheng, Zhi, et al. Monte carlo tree search for comprehensive exploration in llm-based automatic heuristic design. ICML 2025.
>
> [9] Jiang, Xia, et al. Large Language Models as End-to-end Combinatorial Optimization Solvers. NeurIPS 2025.
>
> [10] Queiroga, Eduardo, et al. 10,000 optimal CVRP solutions for testing machine learning based heuristics. AAAI-22 ML4OR Workshop, 2021.
>
>
> *   Additional experiments:
>     To further address your concern, we conducted experiments on newer benchmarks and larger instances, where our results consistently outperform commonly used metaheuristics and existing LLM-driven AHD methods.
>
>     *   Results on new 64 XML CVRP instances [10]:
>
> | Method | Average Cost | Avg Relative Gap (%) |
> | :--- | :--- | :--- |
> | HGS (PyVRP) | 17953.40 | 0.00 |
> | Constructive NN | 22563.79 | 30.42 |
> | Constructive Insert | 23121.25 | 30.59 |
> | OR-Tools GLS | 18429.38 | 2.79 |
> | OR-Tools SA | 18852.62 | 5.70 |
> | OR-Tools TS | 18365.94 | 2.54 |
> | ALNS | 18663.75 | 3.55 |
> | Tabu Search | 18858.68 | 5.72 |
> | GLS EoH | 18537.64 | 2.92 |
> | GLS ReEvo | 18593.80 | 3.23 |
> | GLS MCTS | 18598.92 | 3.26 |
> | Memetic Search | 21105.75 | 15.80 |
> | **MTHS** | **18299.67** | **1.98** |
>
>     *   Results on TSPLib instances of sizes 1000--2000.
>
> | Method | Average Gap |
> | :--- | :--- |
> | Constructive NN | 0.244 |
> | Constructive Insert | 0.204 |
> | OR-Tools SA | 0.108 |
> | OR-Tools TS | 0.105 |
> | ALNS | 0.090 |
> | Tabu Search | 6.493 |
> | Memetic Search | 0.086 |
> | GLS EoH | 0.050 |
> | GLS ReEvo | 0.049 |
> | GLS MCTS | 0.050 |
> | MHTS | **0.035** |

---

> ### Author Response · Authors · 2025-11-20
> **Response to Reviewer MwQv (Part 2/3)**
>
> > **W2: Latest solvers.** I am not up on all the latest in solvers, but it seems to me their comparison points are older, general solution methods. Perhaps it is reasonable to compare against other general metaheuristic methods, but I do not believe they would be competitive with strong heuristics (even if problem-tailored).
>
> First, our goal is not to surpass the best problem-specific solvers for each optimization task (which is practically infeasible). Rather, we aim to investigate how, and to what extent, LLM-driven AHD can design general-purpose metaheuristics across problems, instead of crafting a heuristic for each task. This cross-problem metaheuristic design is significant but has not been studied by prior work.
>
> We focus on general-purpose metaheuristics and select commonly used baselines from the literature [11], showing that automated design can outperform widely used general metaheuristics (which are not problem-tailored). We agree these are not tailored solvers; problem-specific design via LLM-driven AHD has been studied elsewhere [6], [7]. While our framework could incorporate detailed problem-specific design, doing so would be out of scope here and potentially misleading regarding our main objective.
>
> That said, we agree that comparing against strong problem-specific metaheuristics clarifies how far our general-purpose designs are from tailored state of the art (SOTA). We therefore added SOTA metaheuristics for each task: LKH3 for TSP and HGS for CVRP. To maintain focus, we have omitted some of the weaker results in the following summary:
>
> | | TSPLib | | | | CVRPLib | | | | | | |
> | :--- | :--- | :--- | :--- | :--- | :--- | :--- | :--- | :--- | :--- | :--- | :--- |
> | | 50-99 | 100-199 | 200-499 | 500-1000 | A | B | E | F | M | P | X |
> | LKH3 | 0.49 | 0.12 | 0.00 | 0.15 | - | - | - | - | - | - | - |
> | HGS | - | - | - | - | 0.32 | 0.36 | 0.10 | 0.72 | 1.02 | 0.25 | 0.59 |
> | GLS\_EoH | **0.67** | 0.63 | 1.62 | 2.67 | 2.69 | 3.89 | 3.99 | 6.56 | 4.43 | 5.23 | 5.17 |
> | GLS\_ReEvo | 0.79 | 0.68 | 1.71 | 2.72 | 2.60 | 3.72 | 4.00 | 6.96 | **2.45** | 5.61 | 5.62 |
> | GLS\_MCTS | 0.75 | 0.64 | 1.53 | 2.93 | 3.07 | 3.97 | 4.79 | 6.89 | 4.23 | 5.02 | 6.22 |
> | STHS | 0.87 | 0.60 | 1.47 | 3.59 | 3.48 | 3.88 | 4.41 | 3.41 | 6.80 | 5.64 | 5.36 |
> | MTHS | 0.72 | **0.49** | **1.03** | **2.64** | 1.08 | 1.50 | **0.94** | **1.23** | 3.51 | **1.06** | **4.29** |
>
> [11] Gendreau, Michel, et al. Handbook of Metaheuristics. Springer, 2019.
>
> > **W3: Explanation and understanding the system.** There’s little explanation or understanding of why the system works or why it would fail. For example, there’s no study of how the “knowledge transfer” actually helps, or what happens if the tasks are not sufficiently closely related.
>
> Thank you for the valuable comments. We address your concerns as follows:
> *   We present an ablation study to quantify the effect of knowledge transfer. It shows that:
>
> | | TSP | CVRP | FSSP |
> | :--- | :--- | :--- | :--- |
> | MTHS | 0.00564 | 0.04122 | 0.14179 |
> | MTHS w/o Knowledge Transfer | 0.00570 | 0.04446 | 0.13630 |
> | MTHS w/o Knowledge transfer & Low-Level | 0.00636 | 0.04495 | 0.14509 |
>
> *   We added an illustration of the knowledge transfer process in Appendix Figure 7 of the revised manuscript. All three solvers (TSP, CVRP, FSSP) are based on our Adaptive Cooperative Partitioned Evolutionary Refinement (ACPER) framework. Knowledge transfer is implemented as an internal refinement: we upgraded the CVRP and FSSP implementations by adapting specific strategies first validated in our more mature TSP solver. In Figure 7, we detail one such transfer and its impact.
>
> *   We also investigated whether a metaheuristic designed by our framework can enhance the problem-solving capabilities of various LLMs on a new, unseen task. We test two additional tasks that are not combinatorial optimization: black-box optimization problem and admissible set problem. We prompted four different LLMs (GPT-5-mini, Gemini-2.5-pro, Claude-3.7, and GPT-3.5-turbo) to solve the task, both with and without the guidance of the metaheuristic (denoted as `+ MH').
>
>     The following tables report the average of the top-10 algorithms among 100 samples and the best score (lower is better). Results show that:
>     *   The metaheuristic, originally designed for combinatorial optimization, generalizes effectively to the black-box optimization task despite its different structure and settings. We observe notable performance improvements across all four LLMs, regardless of their size and capabilities.
>     *   Conversely, the metaheuristic does not generalize well to the admissible set task (which has quite different settings and features), where it generally provides no improvement. For instance, with GPT-5-mini, both the average and best scores worsened when using the metaheuristic, while only a slight improvement was observed for Gemini-2.5-pro.

---

> ### Author Response · Authors · 2025-11-20
> **Response to Reviewer MwQv (Part 3/3)**
>
> | Method | Average Score | Best Score |
> | :--- | :--- | :--- |
> | GPT-5-mini | 2.8656 | 1.2192 |
> | GPT-5-mini + MH | 0.3254 | 0.0265 |
> | GPT-3.5-turbo | 21.7675 | 14.8752 |
> | GPT-3.5-turbo + MH | 27.8859 | 3.7699 |
> | Claude-3.7-sonnet | 0.5378 | 0.0000 |
> | Claude-3.7-sonnet + MH | 0.4239 | 0.0000 |
> | Gemini-2.5-pro | 10.8157 | 2.9874 |
> | Gemini-2.5-pro + MH | 1.6639 | 0.3035 |
>
> | Method | Average Score | Best Score |
> | :--- | :--- | :--- |
> | GPT-5-mini | 631.5000 | 294.0000 |
> | GPT-5-mini + MH | 899.1000 | 426.0000 |
> | Gemini-2.5-pro | 824.1000 | 684.0000 |
> | Gemini-2.5-pro + MH | 742.6500 | 279.0000 |
>
> > **W4: Efficiency.** The experiment suggests this approach is expensive and slow to run. The paper doesn’t show whether this approach could scale to larger or more diverse problems, or whether it’s practical for anyone to use for more “real-life” problems.
>
> *   The proposed method is more efficient than existing LLM-driven AHD approaches in terms of both runtime and the number of evaluations required because it optimizes multiple problems in a single run. We have compiled a detailed comparison of token consumption, number of evaluations, and wall-clock time against several baseline LLM-driven Automated Heuristic Design (AHD) methods. The results, averaged across the TSP and CVRP tasks, are presented below.
>
> | **Method** | **Tokens (Approx.)** | **# Evaluations** | **Time Cost (Approx.)** |
> | :--- | :--- | :--- | :--- |
> | ***Traveling Salesperson Problem (TSP)*** | | | |
> | EoH | 3.0M | 1000 | 8 h |
> | ReEvo | 3.2M | 1000 | 9 h |
> | MCTS-AHD | 3.1M | 1000 | 40 h |
> | **Ours (MTHS)** | **2.7M** | **333** | **8 h** |
> | ***Capacitated Vehicle Routing Problem (CVRP)*** | | | |
> | EoH | 3.2M | 1000 | 10 h |
> | ReEvo | 3.3M | 1000 | 9 h |
> | MCTS-AHD | 3.5M | 1000 | 45 h |
> | **Ours (MTHS)** | **2.7M** | **333** | **8 h** |
>
> *   As shown in our response to W1, we included additional results on newer benchmarks (CVRP XML set) and larger instances (TSPLib 1000-2000). The results demonstrate the consistent good performance on these instances.
>
>     [1] Queiroga, Eduardo, et al. 10,000 optimal CVRP solutions for testing machine learning based heuristics. AAAI 2021.
>
> *   As shown in point 3 of our response to W3. We have conducted experimental studies on two additional problems that are not combinatorial optimization problems.
>
> > **W5: Generalization.**
>
> Please refer to the responses for W1–W4.
>
> > **Q1&Q2.** I would like to see tests on larger and more modern datasets, and comparisons with current best solvers (even if problem-specific). Is there any type of analysis or insight about how knowledge transfer works here, or other aspects of the system?
>
> We have provided additional results and comparisons to strong problem-specific solvers. As noted, our focus is not to surpass tailored solvers but to study how, and to what extent, LLM-driven AHD can design generalized metaheuristics. We demonstrate competitive performance against commonly used general metaheuristics and against existing LLM-driven AHD methods. Details are in the responses to W1–W3.
>
> We also added ablation studies on knowledge transfer and an example illustrating the transfer process (from TSP to JSSP). We further investigated generalization to different task types. The results show that:
> *   MTHS with knowledge transfer achieves better overall results.
> *   Knowledge transfer enables learning from elite implementations on related tasks, even when tasks differ in constraints and settings. A detailed illustration of a transfer from TSP to FSSP is shown in Figure 7 in the Appendix of the revised manuscript.

---

> > ### Comment · Reviewer_MwQv · 2025-11-25
> >
> > I have seen the responses given by the authors to my and other reviews.  I appreciate their work.  I am planning to keep my score the same.

---

> > > ### Author Response · Authors · 2025-11-26
> > >
> > > Dear Reviewer MwQv,
> > >
> > > Thank you for your time and for acknowledging our responses.
> > >
> > > We would appreciate further clarification regarding the decision to maintain the score of 2 (Reject), as we hoped to have fully addressed the weaknesses and concerns listed in your review with our new experiments and clarification.
> > >
> > > To recap our revision efforts based on your feedback:
> > >
> > > + Regarding "Small/Old Datasets": We conducted new experiments on large-scale instances (up to 2000 nodes) and modern benchmarks (XML CVRP) to demonstrate effective scaling.
> > > + Regarding "Comparison with Solvers": We added comparisons against state-of-the-art, problem-specific solvers (LKH3 for TSP, HGS for CVRP) to show our generalist approach remains competitive.
> > > + Regarding "Understanding the System": We provided a new ablation study to confirm the contribution of the knowledge transfer module, along with a visualization (Figure 7) to explain the mechanism.
> > > + Regarding "Efficiency": We provided a breakdown showing our method uses fewer tokens and less time than leading baselines (EoH, ReEvo).
> > >
> > > As these revisions were made specifically to resolve the grounds for the initial score, we are unsure if there are remaining concerns we might have overlooked.
> > >
> > > If there are other concerns or if the new results did not meet your expectations, could you please provide specific details? We are eager to engage in further discussion to ensure the paper is as strong as possible.
> > >
> > > Sincerely,
> > >
> > > The Authors

---

### Author Response · Authors · 2025-11-20
**General Response**

We sincerely thank the reviewers for their insightful comments and constructive feedback. We have carefully considered all the comments and suggestions, which have significantly improved the manuscript. The major revisions and new results are summarized below:

### 1. Expanded Benchmarks: Larger Scales and SOTA Baselines
**(Reviewers MwQv, wa6u)**

To address concerns regarding dataset size and baseline strength, we have expanded our evaluation across different dimensions:
- **Larger Instances:** We extended experiments to TSPLib instances of sizes 1000--2000 and new CVRP instances (XML instances).
- **SOTA Comparisons:** We added comparisons against state-of-the-art problem-specific metaheuristics (LKH3 for TSP and HGS for CVRP).
- **Result:** Our method (MTHS) remains competitive against general-purpose metaheuristics and existing LLM-driven methods even at these scales. While tailored SOTA solvers (LKH3/HGS) naturally hold an edge, MTHS for the first time demonstrates that LLM-driven AHD can bridge the gap in designing cross-problem generalized metaheuristics.

### 2. Robustness Across Diverse LLMs
**(Reviewers wa6u, dyxi, p6UP)**

We conducted a comprehensive sensitivity analysis using six different LLMs, ranging from commercial SOTA to open-source models: GPT-5-mini, Gemini-2.5-pro, Claude-3.7, GPT-3.5-turbo, Deepseek-V3, and Qwen3, across both design and generalization phases.
- **Design Phase:** We found that while stronger models (GPT-5-mini, Gemini-2.5-pro) yield better metaheuristics, our framework remains effective with less capable models.
- **Generalization Phase:** Crucially, we demonstrate that a metaheuristic designed by MTHS can enhance the problem-solving capabilities of different LLMs (e.g., Claude-3.7, GPT-3.5) on unseen tasks.

### 3. Generalization and Knowledge Transfer
**( Reviewers MwQv, wa6u, dyxi)**

We have conducted additional experiments on two new tasks (continuous black-box optimization and mathematical task) and provided a quantitative and qualitative analysis of the generalization limits:
- **Cross-Domain Success:** We successfully transferred metaheuristics evolved on combinatorial problems (TSP/CVRP/FSSP) to a continuous **Black-Box Optimization** algorithm design task, achieving significant performance gains across multiple LLMs.
- **Defining Boundaries:** We tested on a mathematical task (**Admissible Set**), where the metaheuristic did not improve performance because the problem has totally different settings and features.
- **Mechanism:** We provided a case study illustrating how knowledge is conceptually transformed across tasks (e.g., TSP to FSSP).

### 4. Computational Efficiency and Cost Analysis
**(Reviewers wa6u, dyxi)**

We have added a detailed cost comparison against baseline LLM-driven AHD methods.
- **Efficiency:** MTHS requires approximately **1/3 of the algorithm evaluations** compared to single-task baselines (333 vs. 1000 evaluations per task).
- **Scalability:** Because the high-level metaheuristic is reused, the marginal cost per new task is significantly lower than existing methods, which require independent search runs for every new problem.

### 5. Clarification on Novelty and Related Work
**(Reviewers p6UP, vop9)**

We have clarified the distinction between our work and recent studies like EoH-S and meta learning. While prior works focus on **cross-distribution** generalization (e.g., small to large instances of the same problem), MTHS targets totally different **cross-problem** generalization (e.g., TSP to FSSP). Our hierarchical representation (separating task-agnostic metaheuristics from task-specific implementations) enables this broader transfer, which is not feasible with previous works.

---

### Author Response · Authors · 2025-12-01
**Authors' Summary to Area Chair**

Dear AC,

We sincerely thank you and the reviewers for the time and effort invested in evaluating our work. To assist in your final assessment, we provide a concise summary of our rebuttal and the additional experiments conducted.

While we are encouraged by the positive recognition of our hierarchical framework's novelty (Scores: 6, 6 from Reviewers wa6u, p6UP), we understand that the initial evaluation with lower scores (4, 4, 2 from Reviewers MwQv, dyxi, vop9) stemmed primarily from concerns regarding **experimental scale**, **generalization boundaries**, and **robustness**.

We have taken these critiques seriously and conducted extensive new experiments to address them. We believe the results, **summarized below in One Table** for ease of review, provide strong evidence that resolves the reviewers' main reservations.


| Reviewer Concern | Key Response & Evidence |
| :--- | :--- |
| **1. Dataset Scale & Quality** *(Reviewer MwQv)*  Concern: *Limited scale and benchmark quality compared to SOTA.* | **Expanded Evaluation to 2000 Nodes:** We extended experiments to **TSPLib instances up to 2000 nodes** and the new **XML CVRP benchmark**.  **Comparison to Recent accepted papers:** We now outperform the evaluation scope of recent accepted papers: • **Ours:** Sizes **51–2000** (TSPLib, CVRPLib, Taillard) • **ReEvo (NeurIPS '24):** Sizes 51–1577 • **MCTS-AHD (ICML '25):** Sizes 51–500 • **EoH (ICML '24):** Sizes 51–200 • **LLM-End2End (NeurIPS '25):** Sizes 51–100 |
| **2. Robustness to Different LLMs** *(Reviewers wa6u, dyxi, p6UP)*  Concern: *Reliance on high-end models; generalization to open-source models.* | **Verified Across 6 Models:** We tested training/testing on **Commercial** (GPT-5-mini, Gemini-2.5-pro, Claude-3.7, GPT-3.5-turbo) and **Open-source** (Deepseek-V3, Qwen3) models.  **Key Findings:** 1. Framework functions effectively across **all** tested architectures. 2. **Transferability Confirmed:** A metaheuristic designed by MTHS can guide different LLMs (e.g., Claude-3.7, GPT-3.5) to solve unseen tasks (problems). 3. *Note:* We acknowledge a performance drop on smaller models like GPT-3.5-turbo. |
| **3. Generalization & Transfer** *(Reviewers MwQv, wa6u, dyxi)*  Concern: *Need evidence of "Knowledge Transfer" across boundaries.* | **Validated on New Domains:** We tested the metaheuristics designed for three combinatorial optimization problems on two different **Continuous Black-Box Optimization** and **Mathematical Admissible Set** problems. The method drastically improves the results demonstrating the generalization of our approach (Lower score is better):  **Black-box Optimization (Avg / Best):** • **GPT-5-mini:** 2.87 / 1.22 $\to$ **Ours: 0.33 / 0.03** • **Claude-3.7:** 0.54 / 0.00 $\to$ **Ours: 0.42 / 0.00** • **Gemini-2.5:** 10.82 / 2.99 $\to$ **Ours: 1.66 / 0.30**  **Admissible Set (Avg / Best):** • **Gemini-2.5:** 824.1 / 684.0 $\to$ **Ours: 742.7 / 279.0** |
| **4. Computational Efficiency** *(Reviewers wa6u, dyxi)*  Concern: *Cost of multi-task evolution vs. single-task baselines.* | **Higher Efficiency:** • **Reduced Number of Evaluations:** MTHS requires **~1/3 of the evaluations** (333 vs. 1000) compared to baselines like EoH and ReEvo. (further reductions can be achievable by scaling the number of training problems.) • **Token Efficiency:** The high-level metaheuristic is **reused across tasks** rather than generated from scratch, significantly improving efficiency, thus lowering token consumption. |
| **5. Novelty vs. Existing Work** *(Reviewers p6UP, vop9)*  Concern: *Differentiation from EoH-S and Meta-optimization.* | **First Cross-Problem Framework:** • **Prior Work:** Focuses on *cross-distribution* generalization (e.g., small $\to$ large instances of the *same* problem). • **MTHS (Ours):** Is the **first** to target **cross-problem** generalization (e.g., transferring logic from TSP $\to$ Flow Shop Scheduling). • **Enabling Representation:** Our hierarchical representation is specifically designed to enable this transfer, which is infeasible with prior representations. |

We have addressed the core concerns of scale, robustness, and novelty with concrete, large-scale data and results. Regrettably, the early closure of the discussion period due to the unexpected incident prevented the reviewers from engaging with these updates. Consequently, we rely on the AC to weigh this extensive new evidence, alongside the consensus on our framework's novelty, when making a final recommendation.

Sincerely,

The Authors

---

### Meta-Review · Area_Chair_kmu1 · 2025-12-29

**Summary:**

The paper proposes Multi-Task Hierarchical Search (MTHS), an LLM-guided evolutionary framework designed to co-evolve general-purpose metaheuristics and task-specific program implementations. By utilizing a hierarchical representation and a knowledge transfer mechanism, the authors aim to overcome the task-specificity limitations of previous Automated Heuristic Design (AHD) methods. The reviewers generally appreciated the motivation and the proposed hierarchical framework. Separating general algorithmic logic from task-specific implementation mirrors expert reasoning and addresses a valid gap in the literature. The experiments demonstrate that the method can achieve strong performance on established combinatorial optimization benchmarks, and the appendices provide good detail regarding prompting procedures. Despite the promising methodology, the consensus among reviewers points to significant experimental and comparative shortcomings that place this submission marginally below the acceptance threshold for ICLR 2026.

The primary initial concerns (before rebuttal) driving this decision are:

1. Multiple reviewers (vop9, p6UP) pointed out missing comparisons with critical recent works, such as "EoH-S" and "Generalizable heuristic generation through large language models with meta-optimization." The absence of these baselines weakens the claims of state-of-the-art performance and novelty. Furthermore, Reviewer MwQv noted that comparisons were largely against older general solution methods rather than modern, strong heuristics.

2. There is a significant concern regarding the fairness of the evaluation (dyxi, wa6u). It is unclear if the baselines utilized the same computational budget or base LLMs as MTHS. Given that MTHS involves a complex, multi-task evolutionary process, the lack of a normalized cost analysis (e.g., token usage, runtime) makes it difficult to determine if the performance gains are due to the method's architecture or simply increased computational resource usage.

3. Reviewers noted that the experiments rely heavily on small, dated problem instances (e.g., TSPLib, CVRPLib) (MwQv). There is skepticism about whether the method scales to larger, modern industrial-scale problems.

4. The evaluation lacks robustness checks across different foundation models (wa6u, p6UP, dyxi). It is unclear if the method is specific to the chosen LLM (e.g., GPT-4o family) or if it generalizes to open-source or weaker models. Additionally, the mechanics of the "knowledge transfer" remain under-explored quantitatively; reviewers requested more insight into how task diversity affects generalization.

While the hierarchical approach to AHD is intuitive and potentially impactful, the empirical validation currently leaves too many open questions regarding fairness, scalability, and comparative performance against the latest SOTA. And during the rebuttal period, the authors provide sufficient details on the experimental scale, generalization boundaries, and robustness of the method. However, I still think that this paper is still marginally below the acceptance threshold. But would not mind if paper is accepted.

**Reviewer Concerns:**

The authors provided a strong rebuttal that addressed several key reviewer concerns:

1. Expanded benchmarks to larger instances (TSPLib up to 2000) and included domain-specific SOTA comparisons (LKH3, HGS).

2. Conducted a comprehensive sensitivity analysis across six LLMs (including open-source models), validating the method's stability.

3.  Clarified generalization boundaries through additional tasks (continuous optimization vs. mathematical reasoning).

Despite improvements, the submission remains marginally below the acceptance threshold due to a critical missing validation:

1. Reviewers (p6UP, vop9) explicitly requested empirical comparisons against recent concurrent works (specifically EoH-S). In the rebuttal, the authors argued for a conceptual distinction (cross-problem vs. cross-distribution generalization) but failed to provide the requested experimental head-to-head data.

2. Without these comparisons, it is difficult to verify the paper's claims of superiority over the existing state-of-the-art in LLM-based AHD.

**Reviewer Scores:**

The reviewer MwQv has made the final rating (2: Reject) before the shutdown of rebuttal process. I think the reviewer vop9 and dyxi would maintain the original score (4).

---

### Decision · Program_Chairs · 2026-01-26

Reject